# Dynamics of BMP signaling and distribution during zebrafish dorsal-ventral patterning

**Autumn P Pomreinke[†], Gary H Soh[†], Katherine W Rogers[†], Jennifer K Bergmann, Alexander J Bläßle, Patrick Müller\***

Systems Biology of Development Group, Friedrich Miescher Laboratory of the Max Planck Society, Tübingen, Germany

**Abstract** During vertebrate embryogenesis, dorsal-ventral patterning is controlled by the BMP/Chordin activator/inhibitor system. BMP induces ventral fates, whereas Chordin inhibits BMP signaling on the dorsal side. Several theories can explain how the distributions of BMP and Chordin are regulated to achieve patterning, but the assumptions regarding activator/inhibitor diffusion and stability differ between models. Notably, 'shuttling' models in which the BMP distribution is modulated by a Chordin-mediated increase in BMP diffusivity have gained recent prominence. Here, we directly test five major models by measuring the biophysical properties of fluorescently tagged BMP2b and Chordin in zebrafish embryos. We found that BMP2b and Chordin diffuse and rapidly form extracellular protein gradients, Chordin does not modulate the diffusivity or distribution of BMP2b, and Chordin is not required to establish peak levels of BMP signaling. Our findings challenge current self-regulating reaction-diffusion and shuttling models and provide support for a graded source-sink mechanism underlying zebrafish dorsal-ventral patterning.
DOI: https://doi.org/10.7554/eLife.25861.001

**\*For correspondence:** pmueller@tuebingen.mpg.de

[†]These authors contributed equally to this work

**Competing interests:** The authors declare that no competing interests exist.

## Introduction

The dorsal-ventral axis is one of the earliest coordinate systems established during animal development and divides the embryo into dorsal (back) and ventral (belly) territories. This axis forms under the influence of the BMP/Chordin patterning system. The activator BMP induces the formation of ventral tissues, and BMP signaling is antagonized on the dorsal side by the inhibitor Chordin. There are currently several disparate models that can explain how BMP signaling is restricted to the ventral side (*Ben-Zvi et al., 2008*; *Barkai and Ben-Zvi, 2009*; *Francois et al., 2009*; *Ben-Zvi et al., 2011b*; *Inomata et al., 2013*; *Ramel and Hill, 2013*; *Ben-Zvi et al., 2014*), but the underlying biophysical assumptions have not been fully tested.

In the 'Graded source-sink + mobile BMP model' (Model 1), BMP is produced in a graded, ventrally biased source, and signaling from diffusing BMP is antagonized by binding to its inhibitor Chordin (*Figure 1—figure supplement 1*, *Table 1*). Chordin (Chd) diffuses from a localized source on the opposing dorsal side and therefore provides a 'sink' that inactivates BMP molecules diffusing through the embryo, helping to shape the signaling distribution into a gradient that peaks ventrally. The distributions of *bmp* and *chd* mRNA in developing embryos are consistent with this idea – initially nearly uniform *bmp* expression refines to a ventrally biased gradient over time (*Ramel and Hill, 2013*; *Zinski et al., 2017*), and *chd* expression is restricted to the dorsal region (*Miller-Bertoglio et al., 1997*).

Similar to Model 1, BMP signaling activity in the 'Graded source-sink + immobile BMP model' (Model 2, *Figure 1—figure supplement 1*, *Table 1*) is also restricted by the inhibitor Chordin diffusing from the dorsal side. However, Model 2 assumes that BMP does not diffuse (*Ramel and Hill,*

**eLife digest** Animals start life as clumps of cells that ultimately give rise to complex structures and organs. Over a century of research has revealed a small number of proteins that are crucial for complex structures to form from these clumps, including one protein called BMP. Different levels of BMP instruct cells to give rise to different tissues. In zebrafish, BMP is more abundant on one side of the embryo than the other. This gradient in BMP levels causes different tissues to form at distinct positions and helps coordinate embryo development.

Several theories have been proposed to explain how the BMP gradient is established. They all suggest that a second protein – Chordin – plays an important role in influencing how cells sense the BMP gradient by blocking BMP's activity. However, the exact role of Chordin in the formation of the BMP gradient is disputed. To address this, Pomreinke, Soh, Rogers et al. directly tested five theories of how BMP and Chordin molecules spread through embryos.

The experiments used microscopy to track the movements of fluorescent versions of both molecules in zebrafish embryos. The measurements contradict one theory stating that BMP does not move, and another in which Chordin increases the mobility of BMP. Pomreinke, Soh, Rogers et al. also found that embryos that lack Chordin have increased BMP signaling levels only on the side where Chordin is normally made but not on the opposite side where BMP is made, ruling out several of the theories. The findings are most consistent with the idea that the BMP gradient forms mainly as a result of higher production of BMP on one side of the embryo combined with movement of BMP away from where it is made. Chordin produced at the opposite end of the embryo helps to ensure that only the correct cells receive instructions from BMP.

In the future, two approaches could further clarify how the BMP gradient is formed. First, better techniques to directly observe the BMP gradient in normally developing embryos would be useful. Second, new theories that take into account additional players other than BMP and Chordin might help explain some features of development that current theories cannot address. Uncovering the mechanisms that control the formation of BMP gradients will improve our understanding of how clumps of cells can develop into animals.

DOI: https://doi.org/10.7554/eLife.25861.002

*2013*) and that it binds to Chordin with weaker affinity than in Model 1 (see Materials and methods). Proponents have argued that the similarities between the graded *bmp* mRNA distribution, signaling gradient, and target gene expression indicate negligible BMP diffusion during patterning (*Ramel and Hill, 2013*). Consistent with this, BMP4 was unable to induce long-range signaling in *Xenopus* experiments (*Jones et al., 1996*), although BMP target genes are induced outside of BMP-expressing clones in zebrafish (*Xu et al., 2014*). However, measuring the diffusivity of BMP *in vivo* is the most direct way to determine whether BMP is mobile (*Kicheva et al., 2007*; *Zinski et al., 2017*).

Although these two relatively simple models are generally supported by biological observations, they do not take into account other regulators known to be crucial for dorsal-ventral patterning, such as the BMP-like ligand ADMP, and Sizzled, an inhibitor of the Chordin protease Tolloid/Xlr. Three models described below include these important dorsal-ventral regulators in addition to BMP and Chordin and have also been shown to explain scale-invariant patterning, a phenomenon in which embryos adjust their tissue proportions to differently sized patterning fields.

The recent 'Long-range accumulation and feedback model' (Model 3, *Figure 1—figure supplement 1*, *Table 1*) postulates that BMP and Chordin have equally high mobility, but that dorsal-ventral patterning is controlled by differences in BMP and Chordin protein stability (*Inomata et al., 2013*). In this model, BMP and ADMP induce the secreted, highly diffusible and stable Chordin protease inhibitor Sizzled. This protects Chordin from proteolysis and promotes its expansion towards the ventral side. Over time the resulting inhibition of BMP signaling leads to decreased Sizzled production, destabilizing Chordin and relieving inhibition of BMP. In this way, an appropriate balance between ventral BMP and dorsal Chordin levels can be established even in differently sized embryos.

In the 'Self-regulating reaction-diffusion model' (Model 4, *Figure 1—figure supplement 1*, *Table 1*), BMP and Chordin both have low diffusivities and equivalent protein stabilities. Interactions with highly mobile ADMP and Sizzled in two coupled reaction-diffusion networks eventually result in

**Table 1.** Summary of model assumptions, predictions, and experimental findings.
Model assumptions or predictions that are consistent with the experimental findings (gray) are highlighted in green. NA: no testable model assumptions or predictions.

| | Model 1 Graded source-sink (mobile BMP) | Model 2 Graded source-sink (immobile BMP) | Model 3 Long-range accumulation and feedback | Model 4 Self-regulating reaction-diffusion system | Model 5 Shuttling | Experimental findings |
|---|---|---|---|---|---|---|
| Diffusivity of BMP and Chordin | $D$(BMP) > 0 $D$(BMP) < $D$(Chd) | $D$(BMP) ≈ 0 $D$(Chd) high | $D$(BMP) ≈ $D$(Chd) High | $D$(BMP) ≈ $D$(Chd) Low | $D$(BMP) << $D$(Chd) | $D$(BMP) ≤ $D$(Chd) (≈ 2 and 6 µm$^2$/s) |
| Effect of Chordin on BMP diffusivity | No effect | No effect | No effect | No effect | Chd enhances BMP diffusion | No effect |
| Half-life of BMP and Chordin | τ(BMP) ≈ τ(Chd) | Unconstrained | τ(BMP) >> τ(Chd) | τ(BMP) ≈ τ(Chd) | τ(BMP) > τ(Chd)* | τ(BMP) ≈ τ(Chd) (130 and 120 min) |
| pSmad gradient formation kinetics | Progressive rise ventrally, always low dorsally | Progressive rise ventrally, always low dorsally | Initially high dorsally and ventrally | Progressive rise ventrally, always low dorsally | Progressive rise ventrally, always low dorsally | Progressive rise ventrally, always low dorsally |
| Ventral pSmad peak decreased in the absence of Chordin? | No | No | No | No | Yes | No |
| Dorso-lateral pSmad expansion in the absence of Chordin? | Yes | Yes | Yes | No | Yes | Yes |
| Diffusivity of Sizzled relative to BMP/Chordin | NA | NA | $D$(ADMP) & $D$(Szl) ≈ $D$(BMP) & $D$(Chd) | $D$(ADMP) & $D$(Szl) >> $D$(BMP) & $D$(Chd) | NA | $D$(Szl) ≈ $D$(BMP) & $D$(Chd) (≈ 10, 2, and 6 µm$^2$/s) |

*The simplified shuttling model without ADMP presented here is based on the experimentally measured clearance rate constants of BMP and Chordin; the full model for scale-invariant patterning including ADMP (**Ben-Zvi et al., 2008**) assumes a lower stability of Chordin due to Xlr-mediated degradation.
DOI: https://doi.org/10.7554/eLife.25861.003

the restriction of BMP signaling activity on the ventral side, assuming an initial dorsal Chordin or ventral BMP bias (**Francois et al., 2009**). Such a system self-regulates even with noisy initial conditions and could provide robustness during embryogenesis – e.g., the ability of developing organisms to withstand noise in gene expression or fluctuating environmental conditions – that can be difficult to explain with other models.

Finally, the prominent 'Shuttling model' (Model 5, **Figure 1—figure supplement 1**, **Table 1**) postulates that Chordin not only acts as an inhibitor of BMP, but also modulates the mobility and distribution of BMP protein (**Ben-Zvi et al., 2008**; **Barkai and Ben-Zvi, 2009**; **Ben-Zvi et al., 2011b**; **Ben-Zvi et al., 2014**). In this model, BMP is poorly diffusive, Chordin is highly diffusive, and BMP mobility increases when bound to Chordin. Cleavage of the BMP/Chordin complex by the uniformly distributed protease Tolloid/Xlr combined with a flux of Chordin from the dorsal side is thought to 'shuttle' BMP towards the ventral side by facilitated diffusion over time. In this way, Chordin is responsible for the accumulation of BMP protein on the ventral side, and actively helps establish the subsequent ventral BMP signaling peak.

These five conflicting models postulate different diffusion (no diffusion, equal diffusion, differential diffusion, facilitated diffusion) and stability properties of BMP and Chordin proteins (**Table 1**, **Figure 1—figure supplement 1**). However, these biophysical properties have not been fully measured experimentally, in part due to the lack of reagents and techniques to detect active BMP and Chordin in living vertebrate embryos. To test the biophysical tenets of these models, we developed active BMP and Chordin fluorescent fusion proteins, and used a combination of mathematical modeling and quantitative experiments to determine how BMP2b and Chordin gradients form. Additionally, we tested the distinct predictions that the five models make about how BMP signaling changes in the absence of Chordin. We found that (i) BMP2b and Chordin proteins have similar stabilities, (ii) both BMP2b and Chordin diffuse and form gradients in the extracellular space, and (iii) Chordin does not significantly facilitate BMP2b diffusion or play an active role in establishing peak ventral BMP signaling levels. Together, our results are most consistent with dorsal-ventral patterning mediated by Model 1, the 'Graded source-sink + mobile BMP' model.

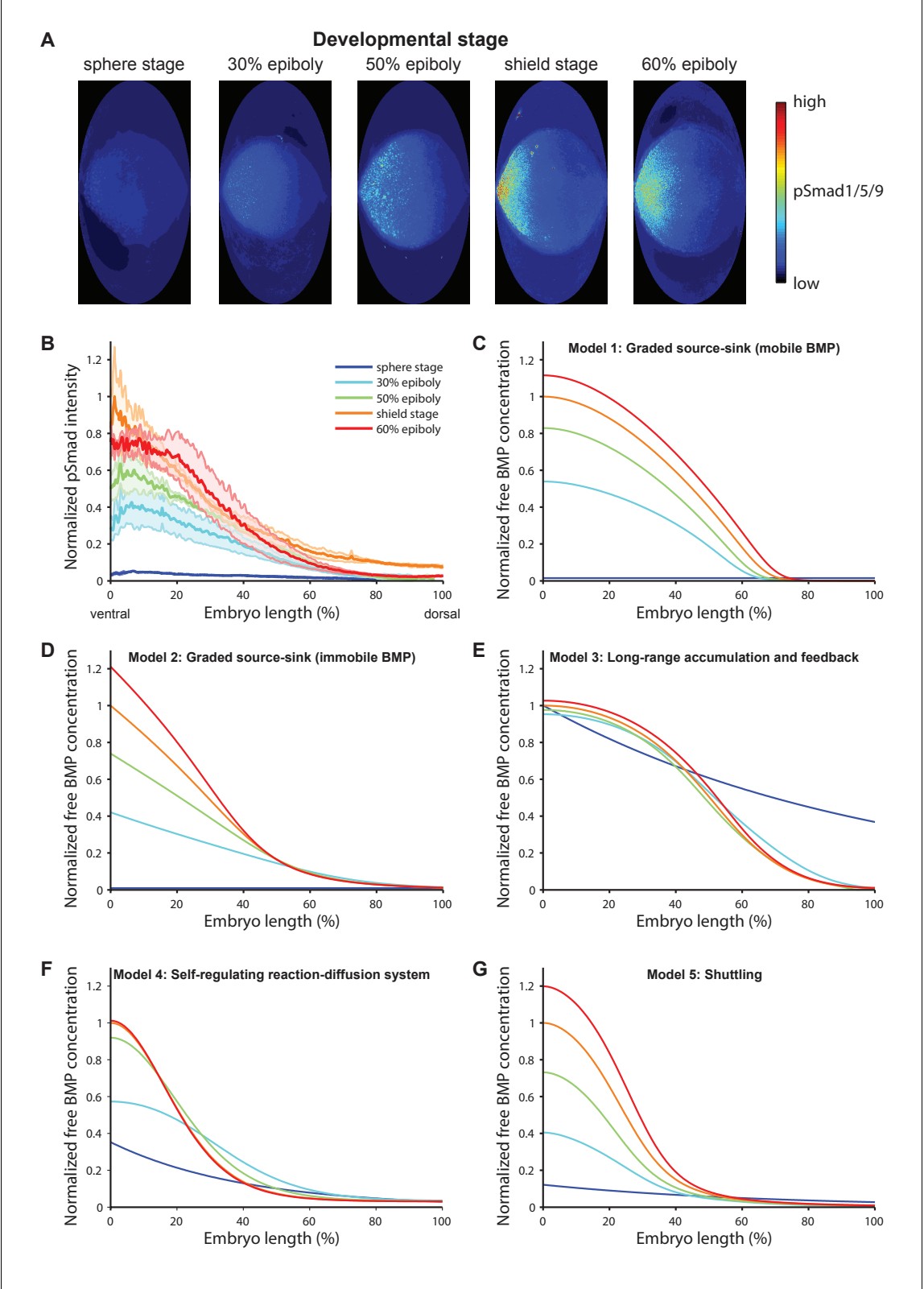

**Figure 1.** BMP signaling (pSmad1/5/9) gradient formation and simulations of five major dorsal-ventral patterning models over relevant zebrafish developmental stages (3 hr). (**A**) Two-dimensional Hammer-Aitoff projections (2D maps) of pSmad1/5/9-immunostained individual wild type zebrafish embryos at different developmental stages. Embryos were imaged using light sheet microscopy (see Materials and methods for details). (**B**) Quantification of ventral-to-dorsal average pSmad1/5/9 distributions in one-dimensional projections of 2D maps generated for embryos at different

*Figure 1 continued on next page*

*Figure 1 continued*

developmental stages (n = 3 for each stage) as in (**A**). Error bars denote standard error. (**C–G**) Gradient formation kinetics simulated for Models 1–5 at relevant zebrafish developmental stages.

DOI: https://doi.org/10.7554/eLife.25861.004

The following figure supplement is available for figure 1:

**Figure supplement 1.** Mathematical formulation of five major models of BMP/Chordin-mediated dorsal-ventral patterning.

DOI: https://doi.org/10.7554/eLife.25861.005

## Results

### Chordin does not actively establish peak ventral BMP signaling

BMP signaling induces phosphorylation and nuclear localization of the transcriptional effectors Smad1/5/9 (*Schier and Talbot, 2005*). To quantitatively measure BMP signaling activity during early dorsal-ventral patterning, we imaged pSmad1/5/9-immunostained zebrafish embryos fixed at different developmental stages using *in toto* light sheet microscopy, converted pSmad1/5/9 signaling activities into information-compressed two-dimensional maps (*Schmid et al., 2013*), and quantified pSmad1/5/9 intensities along the ventral-dorsal axis (*Figure 1A*, Materials and methods). Over the course of approximately 3 hr during early zebrafish development, BMP signaling rapidly shifts from a low-level near-uniform distribution to a gradient with peak levels on the ventral side (*Figure 1A+B*, *Videos 1–5*) (*Tucker et al., 2008*), similar to changes in the distribution of *bmp2b* mRNA over time (*Ramel and Hill, 2013*; *Zinski et al., 2017*). We simulated pSmad1/5/9 gradient formation kinetics predicted by each of the five models over a similar time period (*Figure 1C–G*). Our measurements are consistent with the gradient kinetics predicted by Models 1, 2, 4, and 5, whereas the dynamics predicted by Model 3 do not resemble the experimentally observed distributions.

All five major models of BMP/Chordin-mediated dorsal-ventral patterning qualitatively explain the formation of a ventral signaling peak, but they assign different roles to the inhibitor Chordin (*Figure 2A–E*, *Table 1*, and *Figure 1—figure supplement 1*). Models 1 and 2 assume that a flux of the inhibitor Chordin from the dorsal side restricts the range of BMP signaling activity throughout the embryo. They thus predict that in the absence of Chordin, BMP signaling should be expanded throughout the embryo with a small increase in the peak levels on the ventral side (*Figure 2A+B*). Model 3 adds an additional regulatory layer: Here, the abundance of Chordin is regulated by feedback interactions that modify its stability and affect ventral BMP signaling levels (*Figure 1—figure supplement 1*). Similar to Models 1 and 2, Model 3 also predicts that in the absence of Chordin, BMP signaling should be expanded throughout the embryo (*Figure 2C*).

In Model 4, two reaction-diffusion systems involving BMP/Sizzled and Chordin/ADMP are coupled. In a completely homogenous field of cells with no initial expression biases, this self-organizing system would give rise to both ventral and dorsal BMP peaks (*Francois et al., 2009*). To achieve a single ventral BMP peak, an initial dorsal Chordin or ventral BMP bias is required (see Materials and methods). Under these conditions, the initial advantage in BMP signaling on the ventral side is amplified by autoregulation of BMP production. Since Chordin inhibits the autoregulation of BMP production, the absence of Chordin leads to a more pronounced ventral BMP peak but has no effect in the rest of the embryo (*Figure 2D*). Model 4 thus predicts that in the absence of Chordin, pSmad1/5/9 levels would be increased on the ventral but not the dorsal side.

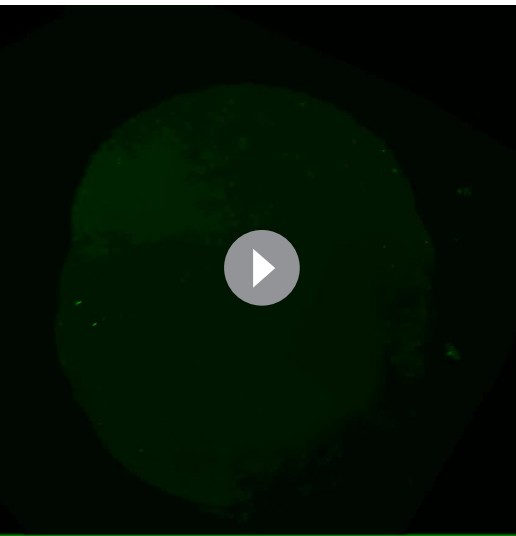

**Video 1.** 3D reconstruction of pSmad1/5/9 localization in a wild type sphere stage zebrafish embryo imaged by light sheet microscopy.

DOI: https://doi.org/10.7554/eLife.25861.006

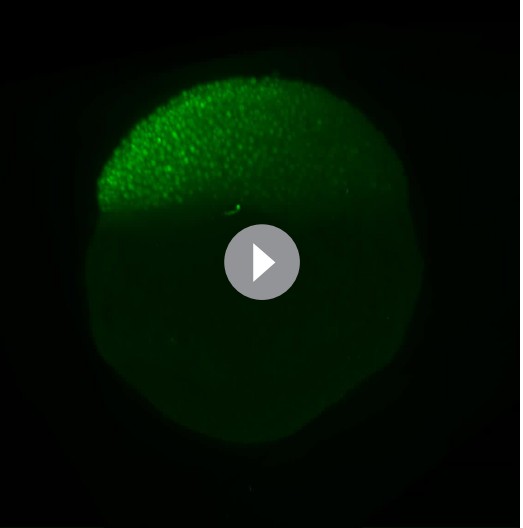

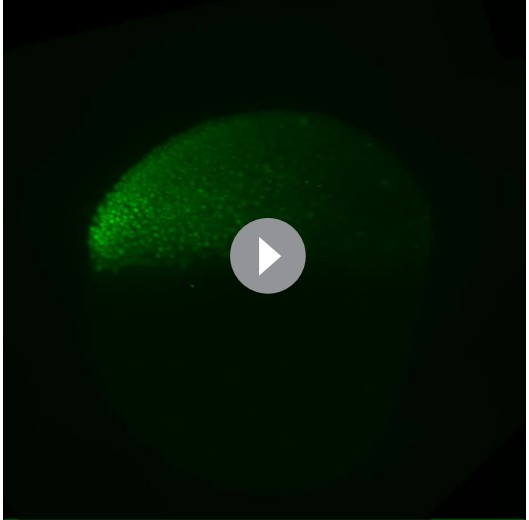

**Video 2.** 3D reconstruction of pSmad1/5/9 localization in a wild type 30% epiboly stage zebrafish embryo imaged by light sheet microscopy.
DOI: https://doi.org/10.7554/eLife.25861.007

**Video 3.** 3D reconstruction of pSmad1/5/9 localization in a wild type 50% epiboly stage zebrafish embryo imaged by light sheet microscopy.
DOI: https://doi.org/10.7554/eLife.25861.008

In contrast to Models 1–4, Model 5 assigns a more active role to Chordin in promoting the ventral BMP signaling peak. This model proposes that Chordin activity results in *increased* BMP signaling ventrally: Chordin increases ventral BMP levels by binding to and physically moving BMP protein towards the ventral side. This model therefore predicts that in embryos lacking Chordin, BMP signaling should be lower on the ventral side compared to wild type embryos (*Figure 2E*).

To experimentally test these predictions, we quantitatively measured BMP signaling activity in fixed *chordin*⁻/⁻ zebrafish embryos (*Video 6*) and their wild type siblings using pSmad1/5/9 immunostaining and *in toto* light sheet microscopy. Strikingly, BMP signaling was increased in dorso-lateral domains in *chordin*⁻/⁻ mutants compared to wild type embryos, but BMP signaling on the

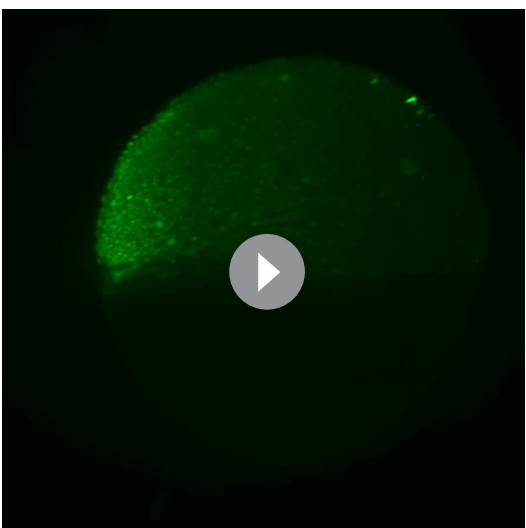

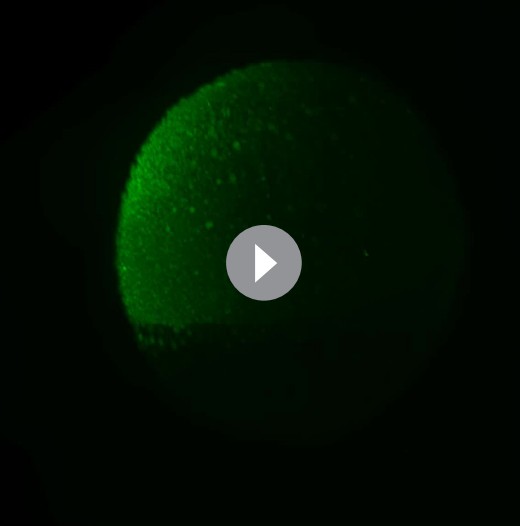

**Video 4.** 3D reconstruction of pSmad1/5/9 localization in a wild type shield stage zebrafish embryo imaged by light sheet microscopy.
DOI: https://doi.org/10.7554/eLife.25861.009

**Video 5.** 3D reconstruction of pSmad1/5/9 localization in a wild type 60% epiboly stage zebrafish embryo imaged by light sheet microscopy.
DOI: https://doi.org/10.7554/eLife.25861.010

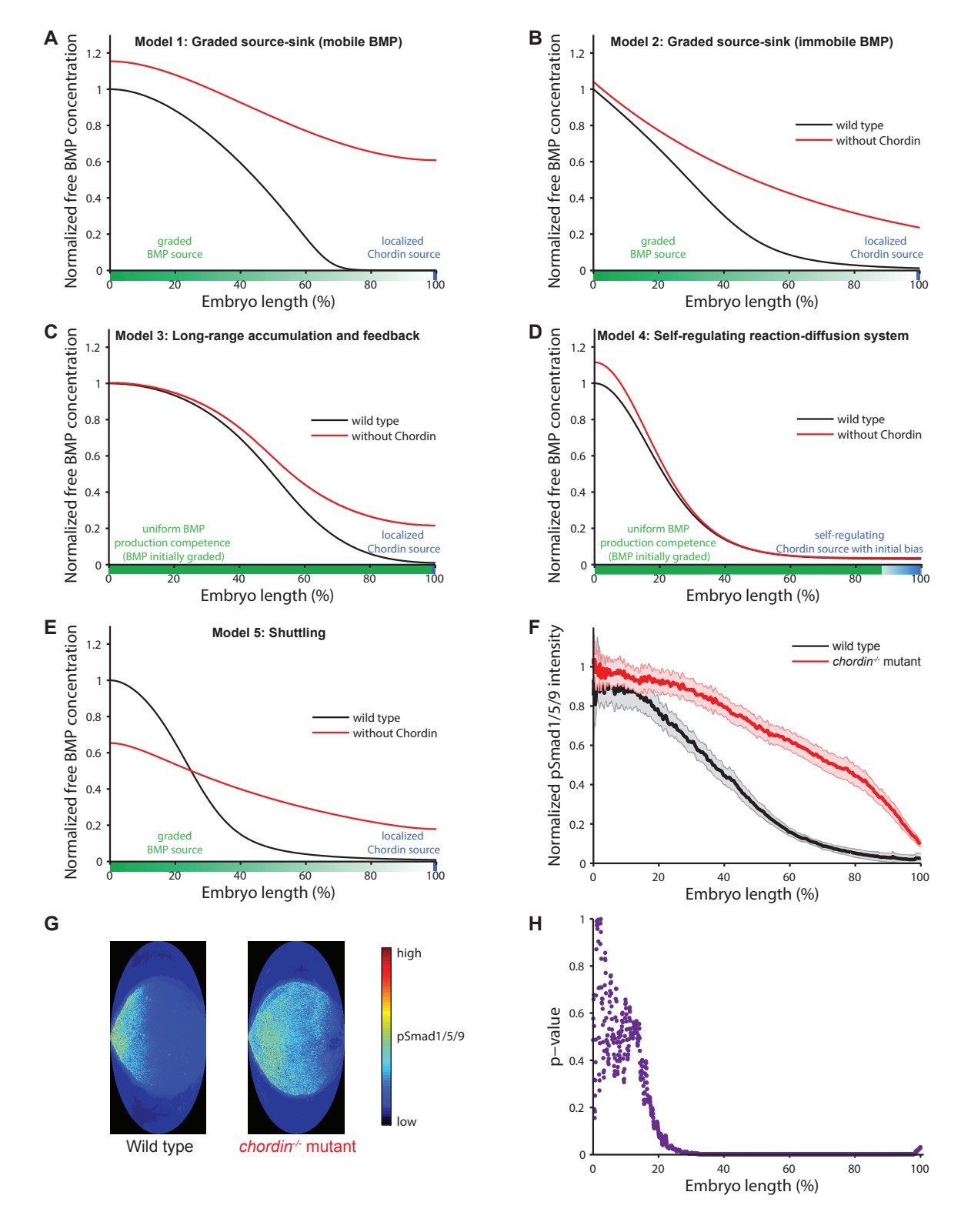

**Figure 2.** Theoretical predictions for the influence of the inhibitor Chordin on the BMP signaling gradient and experimental test. (A–E) Simulations of BMP distributions in five major models of dorsal-ventral patterning in the presence (black) or absence (red) of Chordin. The BMP and Chordin sources are indicated below each graph in green and blue, respectively. Note that the spatial production rates in Models 3 and 4 are modulated over time by feedback. (F–G) Quantification of average pSmad1/5/9 distributions in wild type (black) and *chordin*[−/−] (red) embryos using one-dimensional

*Figure 2 continued on next page*

*Figure 2 continued*

projections of 2D maps. Wild type n = 7, *chordin$^{-/-}$* mutants n = 10. Error bars denote standard error. (H) p-values (unpaired two-tailed t-test assuming equal variance) calculated as a function of space between pSmad1/5/9 distributions in wild type and *chordin$^{-/-}$* embryos shown in (F) indicate no significant difference of pSmad1/5/9 on the ventral side but a dramatic expansion into dorsal-lateral domains.

DOI: https://doi.org/10.7554/eLife.25861.011

ventral side was not significantly affected (*Figure 2F–H*), consistent with the predictions from Models 1–3 and observations in *Xenopus* and zebrafish embryos (*Plouhinec et al., 2013*; *Zinski et al., 2017*), but not with the BMP signaling distributions predicted by Models 4 and 5 (*Table 1*).

## BMP and Chordin fluorescent fusion proteins diffuse and rapidly form gradients *in vivo*

In order to understand the underlying basis of BMP/Chordin distribution and directly test the biophysical assumptions of the five dorsal-ventral patterning models, we developed fluorescent fusion proteins. We fused superfolder-GFP (sfGFP [*Pédelacq et al., 2006*]) and the photoconvertible protein Dendra2 (*Gurskaya et al., 2006*) to zebrafish Chordin and BMP2b, the major BMP ligand regulating zebrafish dorsal-ventral patterning (*Kishimoto et al., 1997*; *Xu et al., 2014*). Basing our design on previously established fusions with small peptide tags (*Cui et al., 1998*; *Degnin et al., 2004*; *Sopory et al., 2006*), we inserted fluorescent proteins to label the mature signaling domains, and obtained fusion proteins that are processed similarly and have similar biological activity as untagged versions or constructs fused to small FLAG tags (*Figure 3A–E*, *Figure 3—figure supplement 1*). Indeed, *BMP2b* mutants (*swr$^{-/-}$*, which are normally severely dorsalized [*Kishimoto et al., 1997*]) can be rescued by injection of mRNA encoding BMP2b-Dendra2 or BMP2b-sfGFP at levels equivalent to untagged BMP2b (*Figure 3C*). In these experiments, the injected mRNA should be uniformly distributed, highlighting the important role of Chordin or other antagonists in shaping the graded BMP signaling distribution.

To measure the kinetics of BMP and Chordin protein gradient formation, we expressed BMP2b-sfGFP and Chordin-sfGFP from local sources in wild type zebrafish embryos (*Müller et al., 2012*) and imaged the distribution profiles over time using light sheet microscopy (*Figure 3F–I*). Importantly, in previous experiments it has been demonstrated that BMP2b clones generated in a similar manner can recapitulate BMP signaling comparable to that observed along the dorsal-ventral axis (*Xu et al., 2014*). Strikingly, both BMP2b-sfGFP and Chordin-sfGFP are secreted and diffuse in the extracellular space (*Figure 3F+G*, *Videos 7+8*), in contrast to the proposal of Model 2 that only Chordin – but not BMP – diffuses (*Ramel and Hill, 2013*) (*Table 1*) and the absence of long-range BMP4 signaling in *Xenopus* (*Jones et al., 1996*). Both BMP2b-sfGFP and Chordin-sfGFP rapidly establish concentration gradients over the course of one hour (*Figure 3H+I*), consistent with the rapid patterning of the dorsal-ventral axis during zebrafish development.

## BMP and Chordin fluorescent fusion proteins have similar stabilities *in vivo*

The gradient formed by Chordin-sfGFP has a moderately longer range than the one formed by BMP2b-sfGFP. For example, 60 min post-transplantation the BMP2b-sfGFP signal drops to 50% of the maximal concentration at a distance of 30–40 µm, whereas the gradient formed by Chordin-sfGFP reaches 50% of its maximal concentration at a distance of 50–60 µm from the source boundary at this time point (*Figure 3H+I*). This

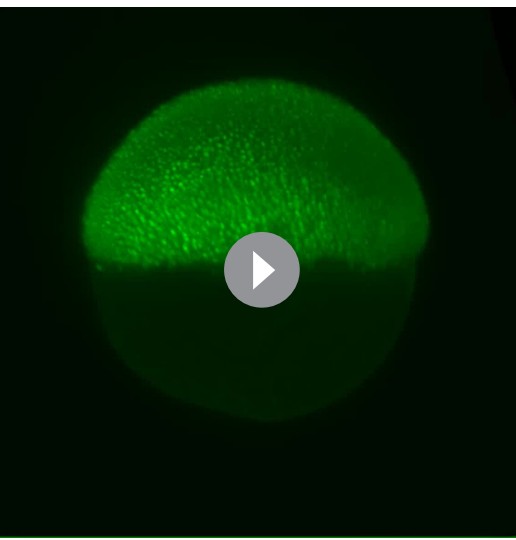

**Video 6.** 3D reconstruction of pSmad1/5/9 localization in a *chordin$^{-/-}$* shield stage zebrafish embryo imaged by light sheet microscopy.

DOI: https://doi.org/10.7554/eLife.25861.012

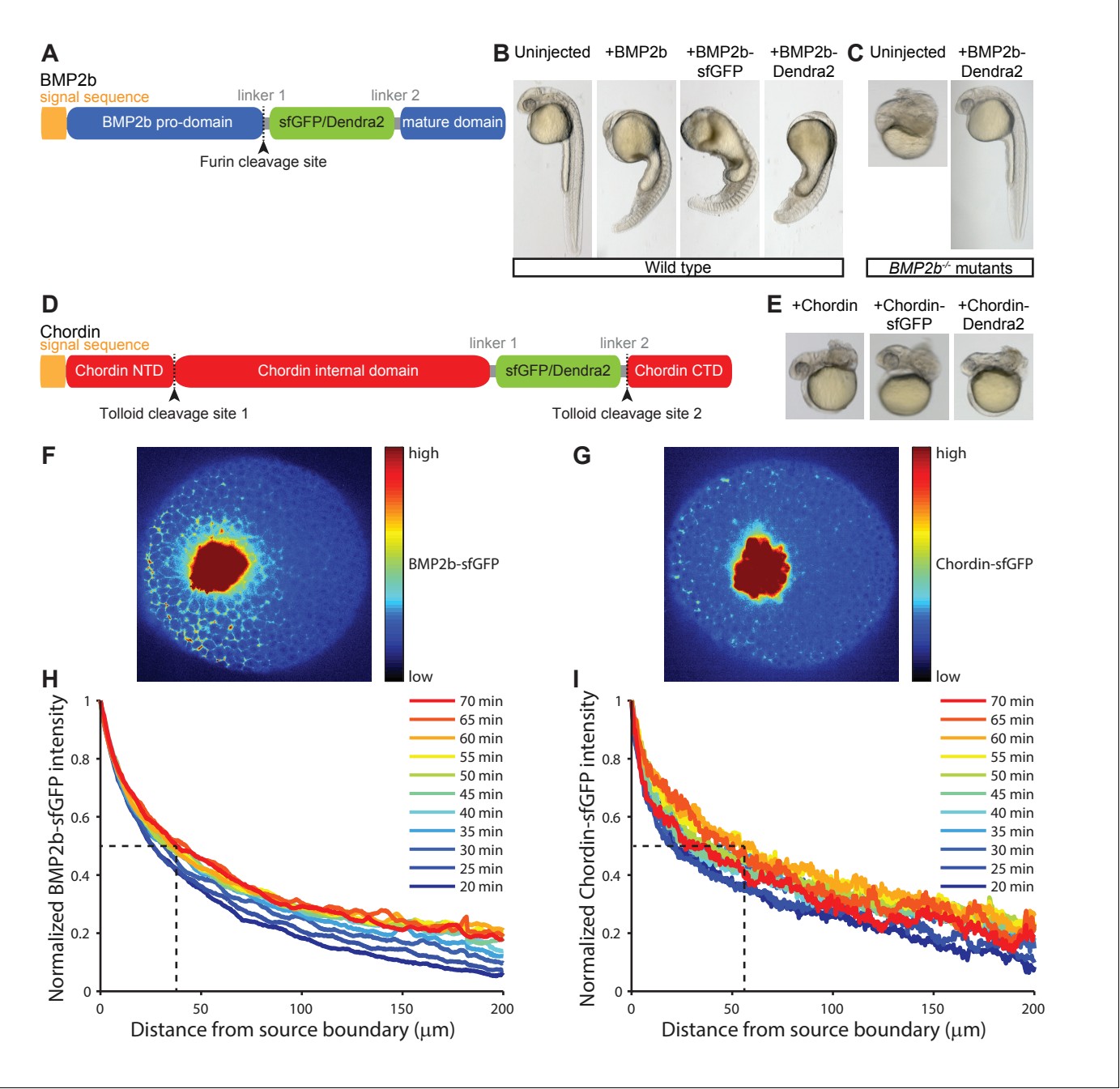

**Figure 3.** Gradient formation kinetics of fluorescently tagged BMP and Chordin. (**A**) Schematic of BMP2b-sfGFP and -Dendra2 fusion constructs. (**B**) Fluorescent BMP2b fusion constructs can induce ventralization, a BMP-overexpression phenotype (*Kishimoto et al., 1997*). mRNA amounts equimolar to 2 pg of *BMP2b* mRNA were injected at the one-cell stage, and images were taken 30 hr post-fertilization (hpf). (**C**) Rescue of a *BMP2b* mutant (*swr*$^{-/-}$) with BMP2b-Dendra2. 2.74 pg of BMP2b-Dendra2-encoding mRNA were injected at the one-cell stage, and images were taken at 30 hpf. In a separate experiment with 1 pg of BMP2b-sfGFP-encoding mRNA, 20% (9/44) of all injected *swr*$^{-/-}$ mutants were rescued, 16% (7/44) were ventralized, and 64% (28/44) were dorsalized. (**D**) Schematic of Chordin-sfGFP and -Dendra2 fusion constructs. (**E**) Fluorescent Chordin constructs can induce dorsalization, a *Chordin*-overexpression phenotype. mRNA amounts equimolar to 30 pg of *Chordin* mRNA were injected into wild type embryos at the one-cell stage, and images were taken at 30 hpf. F + G) Light sheet microscopy images of BMP- and Chordin-sfGFP gradients forming from a local source in live zebrafish embryos. Approximately 50–75 cells expressing *BMP2b-sfGFP* (**F**) or *Chordin-sfGFP* (**G**) were transplanted into host embryos at sphere stage (see Materials and methods for details). The images show gradient formation in single optical slices approximately 20 min after transplantation. H + I) Quantification of BMP2b-sfGFP (**H**) and Chordin-sfGFP (**I**) gradient formation kinetics from a local source (BMP2b-sfGFP: n = 8; Chordin-sfGFP: n = 5). Dashed lines indicate the distance at which the protein distributions drop to 50% of their maximal concentration 60 min post-transplantation.

*Figure 3 continued on next page*

*Figure 3 continued*

DOI: https://doi.org/10.7554/eLife.25861.013

The following figure supplements are available for figure 3:

**Figure supplement 1.** Detailed characterization of fluorescently tagged BMP2b and Chordin.

DOI: https://doi.org/10.7554/eLife.25861.014

**Figure supplement 2.** Modeling of BMP and Chordin gradient formation kinetics and comparison to measured gradients.

DOI: https://doi.org/10.7554/eLife.25861.015

suggests that stability or diffusivity might differ between these proteins (*Müller and Schier, 2011*; *Müller et al., 2013*). Importantly, Models 3 and 5 assume that BMP is more stable than Chordin, whereas the other models assume either similar or unconstrained stabilities (*Table 1*).

To distinguish between these possibilities, we first determined protein stability in living zebrafish embryos using a Fluorescence Decrease After Photoconversion (FDAP) assay (*Müller et al., 2012*; *Bläßle and Müller, 2015*; *Rogers et al., 2015*). We expressed BMP2b and Chordin fused to the green-to-red photoconvertible protein Dendra2 uniformly in zebrafish embryos, used brief UV exposure to convert the signal from green to red to generate a pulsed protein pool, and monitored the decrease in extracellular red fluorescence over time (*Figure 4A+B*). For BMP2b-Dendra2, we found a clearance rate constant of $k_1 = (8.9 \pm 0.1) \times 10^{-5}$/s (half-life 130 min, *Figure 4A*). For Chordin-Dendra2, we measured a similar clearance rate constant of $k_1 = (9.6 \pm 0.3) \times 10^{-5}$/s (half-life 120 min, *Figure 4B*). The similar clearance rate constants suggest that differential protein stabilities cannot account for the different protein distributions of BMP2b and Chordin. Importantly, these results are inconsistent with the differential protein stabilities predicted by Models 3 and 5 (*Table 1*).

## Diffusivity of BMP and Chordin fluorescent fusion proteins *in vivo*

Our finding that BMP2b- and Chordin-Dendra2 fusions have similar stabilities (*Figure 4A+B*) suggests that differences in diffusivity could account for the slight differences in gradient formation kinetics. Indeed, when we fitted a gradient formation model based on local production, uniform diffusion, and clearance constrained with our measured protein half-lives in a realistic three-dimensional zebrafish embryo-like geometry (*Müller et al., 2012*) to the measured protein distributions, we obtained the best agreement between model and data with lower diffusivity of BMP2b (4 μm²/s) compared to Chordin (6 μm²/s) (*Figure 3—figure supplement 2A+B*).

Importantly, the five models assume distinct BMP and Chordin diffusion properties (*Table 1*, *Figure 1—figure supplement 1*), from no BMP diffusion (Model 2) to substantially higher Chordin

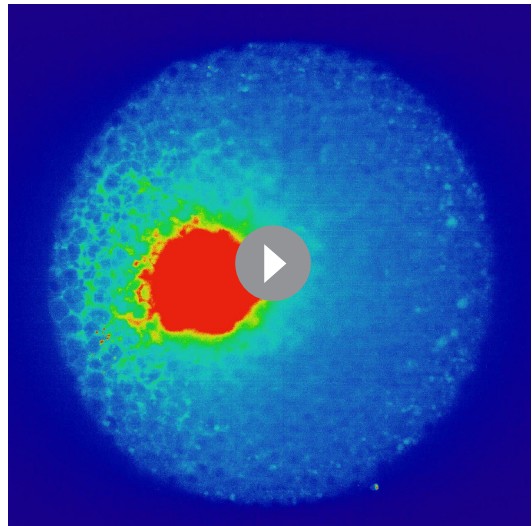 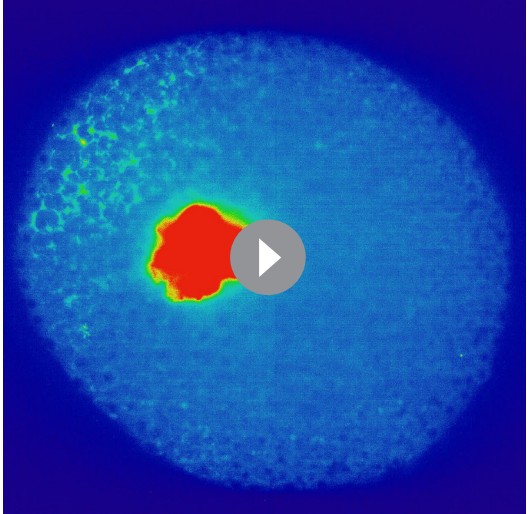

**Video 7.** Gradient formation in a dome stage wild type embryo with a BMP2b-sfGFP clone.
DOI: https://doi.org/10.7554/eLife.25861.016

**Video 8.** Gradient formation in a dome stage wild type embryo with a Chordin-sfGFP clone.
DOI: https://doi.org/10.7554/eLife.25861.017

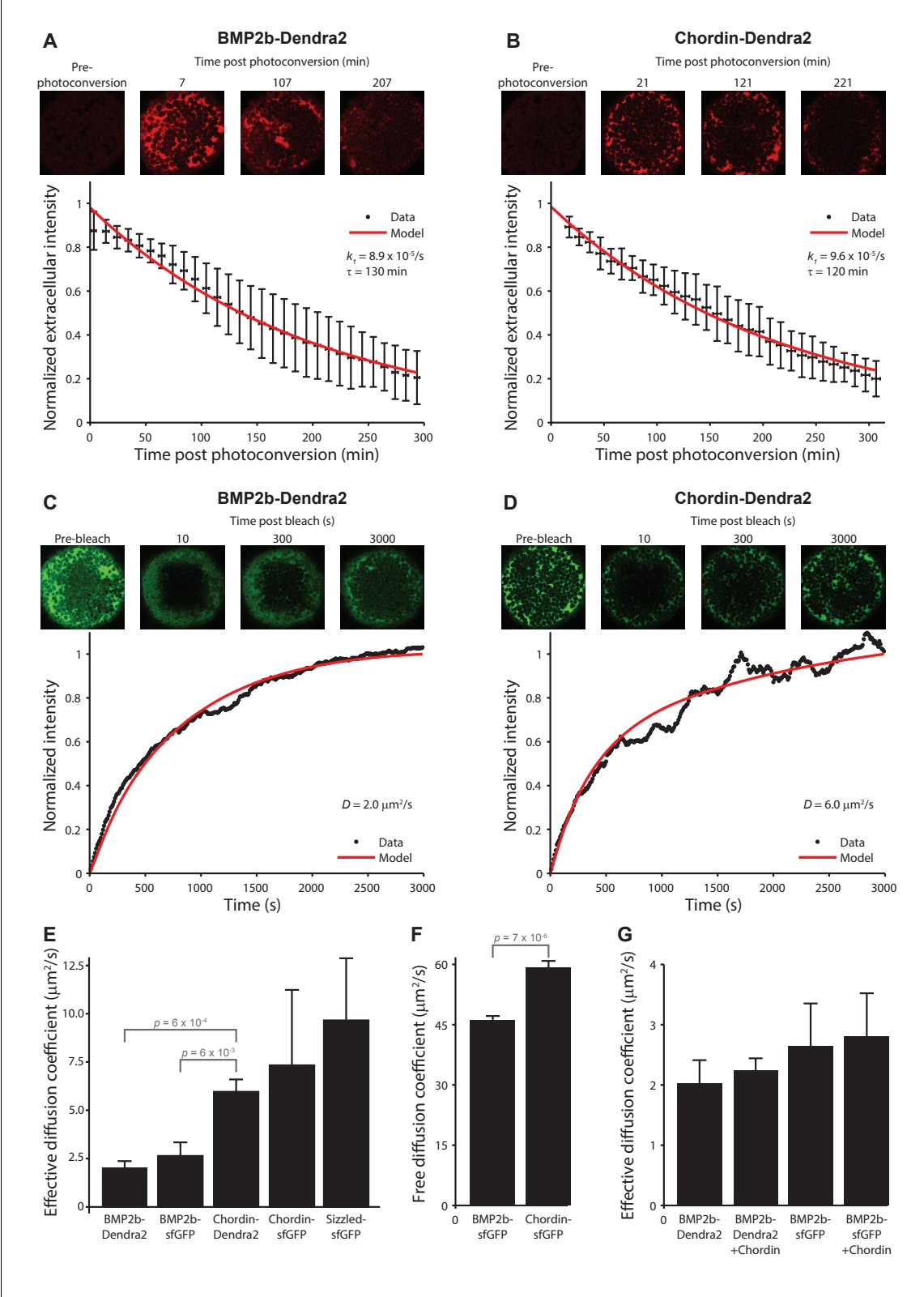

**Figure 4.** Biophysical measurements of BMP and Chordin protein stability and diffusivity. A + B) FDAP protein stability measurements for BMP2b-Dendra2 (A) and Chordin-Dendra2 (B). Error bars denote standard deviation. BMP2b-Dendra2: n = 22; Chordin-Dendra2: n = 6. C + D) FRAP effective protein diffusivity measurements for BMP2b-Dendra2 (C) and Chordin-Dendra2 (D). Data and fits from single experiments are shown. (E) Bar chart of the average effective diffusion coefficients from FRAP experiments. Error bars denote standard error. BMP2b-Dendra2: n = 6; BMP2b-sfGFP: n = 8;
*Figure 4 continued on next page*

*Figure 4 continued*

Chordin-Dendra2: n = 8; Chordin-sfGFP: n = 6; Sizzled-sfGFP: n = 12. (**F**) Free diffusion coefficients of BMP2b-sfGFP and Chordin-sfGFP measured by Fluorescence Correlation Spectroscopy (FCS) in a diffraction-limited spot within the zebrafish embryonic extracellular space far away from cell membranes (see Materials and methods for details). Error bars denote standard error. BMP2b-sfGFP: n = 17 measurements from 4 embryos; Chordin-sfGFP: n = 19 measurements from 5 embryos. (**G**) Negligible influence of Chordin on BMP2b effective diffusion. Untagged Chordin was co-expressed with BMP2b-Dendra2 (n = 8) or BMP2b-sfGFP (n = 9) in zebrafish embryos subjected to FRAP measurements at blastula stages. The data shown for BMP2b-Dendra2 and BMP2b-sfGFP FRAP experiments without co-expressed *Chordin* is identical to the data shown in (**E**). p-values (unpaired two-tailed t-test assuming equal variance) are shown for statistically significant (p<0.05) data sets.

DOI: https://doi.org/10.7554/eLife.25861.018

The following figure supplement is available for figure 4:

**Figure supplement 1.** Characterization of Sizzled diffusion and its role in gradient formation.

DOI: https://doi.org/10.7554/eLife.25861.019

mobility compared to BMP (Model 5). To directly test these predictions, we determined the effective diffusivities of fluorescently tagged BMP2b and Chordin moving through developing zebrafish embryos. We used a Fluorescence Recovery After Photobleaching (FRAP) assay (*Müller et al., 2012*) that measures the dynamics of re-appearance of fluorescence in a bleached region in embryos uniformly expressing fluorescent fusion proteins (*Figure 4C–E*). We found effective diffusion coefficients of 2–3 $\mu m^2$/s for BMPs (BMP2b-Dendra2: 2.0 ± 0.4 $\mu m^2$/s; BMP2b-sfGFP: 2.6 ± 0.7 $\mu m^2$/s (similar to [*Zinski et al., 2017*]) and of 6–7 $\mu m^2$/s for Chordin (Chordin-Dendra2: 6.0 ± 0.7 $\mu m^2$/s; Chordin-sfGFP: 7.3 ± 3.9 $\mu m^2$/s), indicating that slight differences in diffusivities could underlie the differences in protein distributions. This idea is further supported by the agreement between gradients simulated with the measured diffusivities and clearance rate constants and our experimentally determined protein gradients (*Figure 3—figure supplement 2E–H*). The measured diffusion coefficients are most consistent with Models 1 and 4, which assume either similarly low diffusivities (Model 4) or that BMP has a moderately lower diffusion coefficient than Chordin (Model 1, *Table 1*). As observed in the BMP2b-sfGFP gradient formation experiment (*Figure 3F–I*), our FRAP data demonstrate that BMP2b-sfGFP is mobile *in vivo*, inconsistent with Model 2.

Strikingly, local diffusion measurements in very small extracellular volumes far away from cell surfaces using Fluorescence Correlation Spectroscopy (FCS) assays showed that BMP2b-sfGFP (free diffusion coefficient: $D_f$ = 46 ± 1 $\mu m^2$/s) and Chordin-sfGFP (free diffusion coefficient: $D_f$ = 59 ± 2 $\mu m^2$/

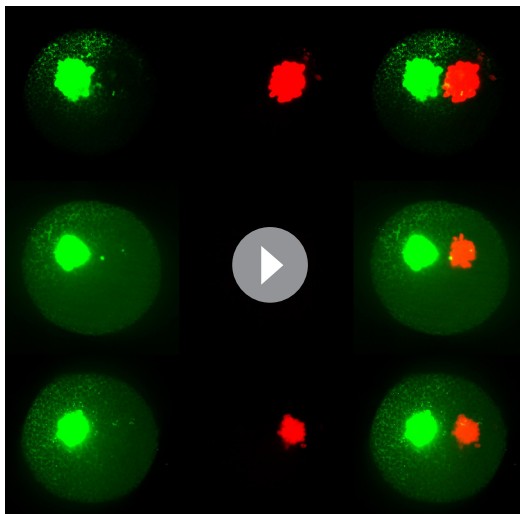

**Video 9.** Gradient formation in three representative dome stage wild type embryos with BMP2b-sfGFP clones (green) next to clones labeled with Alexa 546-coupled dextran (red).

DOI: https://doi.org/10.7554/eLife.25861.021

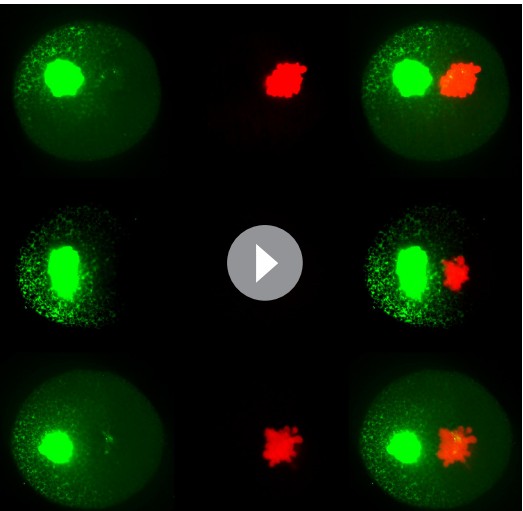

**Video 10.** Gradient formation in three representative dome stage wild type embryos with BMP2b-sfGFP clones (green) next to *chordin*-expressing clones labeled with Alexa 546-coupled dextran (red).

DOI: https://doi.org/10.7554/eLife.25861.022

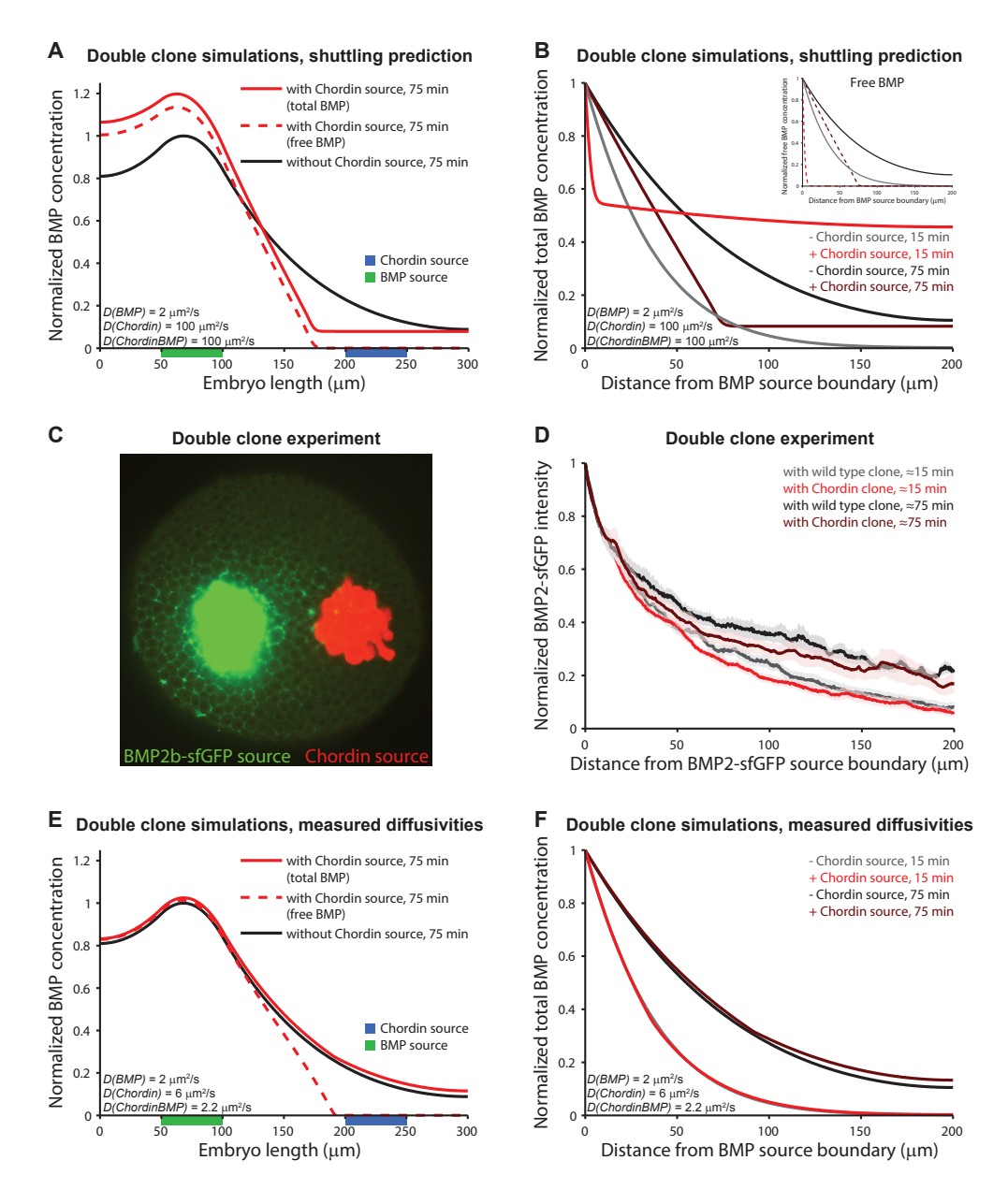

**Figure 5.** Testing shuttling of BMP2b predicted by Model 5. (**A**) One-dimensional model of two clones expressing BMP (green) or Chordin (blue) with $D_{BMP}$ = 2 μm²/s, $D_{Chd}$ = 100 μm²/s, and $D_{ChdBMP}$ = 100 μm²/s. BMP levels increase over time due to constant production. In the presence of Chordin, the BMP gradient is deflected away from the Chordin source indicative of shuttling (compare black and red lines). Solid lines show total BMP levels (i.e. BMP + ChdBMP in the presence of Chordin), and dashed line shows free BMP levels. (**B**) BMP gradients to the right of the BMP-expressing clone re-normalized to the BMP concentration at the source boundary to demonstrate that the range of BMP is decreased between the two clones in the presence of Chordin. The main panel shows total BMP levels (i.e., BMP + ChdBMP in the presence of Chordin), and the inset shows free BMP levels (dashed lines). (**C**) Experimental test of the predictions in (**A**) and (**B**). Clones of cells expressing BMP2b-sfGFP (green) were generated by transplanting approximately 50–75 cells from a donor embryo into wild type hosts at sphere stage (see Materials and methods for details). Another clone of cells (red) was transplanted next to the BMP2b-sfGFP-expressing clone shortly after. The red clone is marked by the presence of fluorescent Alexa 546-coupled dextran. Cells from red-labeled clones either contained only Alexa 546-coupled dextran (*Video 9*) or Alexa-546-coupled dextran and ectopic *chordin* mRNA (*Video 10*). 15–20 min after transplantation of the clones, embryos were imaged using light sheet microscopy. The image shows gradient formation in a single optical slice approximately 20 min after transplantation. (**D**) Quantification of average BMP2b-sfGFP gradients at ~15 min or ~75 min after transplantation in embryos generated as in (**C**) with (red/brown) or without (black/gray) ectopic Chordin sources. Error bars denote standard error. n = 8 for each condition. (**E**) One-dimensional simulation of two clones expressing BMP (green) or Chordin (blue) with the experimentally measured diffusion coefficients $D_{BMP}$ = 2 μm²/s, $D_{Chd}$ = 6 μm²/s, and $D_{ChdBMP}$ = 2.2 μm²/s. BMP levels increase over time due to constant production.
*Figure 5 continued on next page*

*Figure 5 continued*

Solid lines show total BMP levels (i.e. BMP + ChdBMP in the presence of Chordin), and the dashed line shows free BMP levels. Only the distribution of free BMP is affected as a consequence of Chordin binding, and the gradient of total BMP is not deflected away from the Chordin source (compare solid black and red lines). (F) Gradients of total BMP levels to the right of the BMP expressing clone simulated with the experimentally measured diffusion coefficients ($D_{BMP}$ = 2 µm$^2$/s, $D_{Chd}$ = 6 µm$^2$/s, and $D_{ChdBMP}$ = 2.2 µm$^2$/s) and renormalized to the concentration at the boundary show that the range of BMP is not decreased between the two clones in the presence of Chordin.

DOI: https://doi.org/10.7554/eLife.25861.020

s) are highly mobile over short spatial and temporal scales (*Figure 4F*), whereas their diffusivities are reduced at the global scale when they move across a field of cells (*Figure 4E*). We hypothesize that the difference between effective diffusivities (measured by FRAP) and local diffusivities (measured by FCS) is due to binding to immobile extracellular molecules, which could serve as diffusion regulators that hinder the mobility of BMP2b and Chordin, similar to what has been proposed for other developmental signals such as Nodal and FGF (*Müller et al., 2012*; *Müller et al., 2013*).

## Sizzled, BMP, and Chordin diffusivities are within the same order of magnitude

Models 3 and 4 assign important roles to the secreted proteins ADMP and Sizzled in regulating BMP signaling and distribution. Model 3 postulates diffusivities of ADMP and Sizzled equivalent to BMP and Chordin, whereas Model 4 requires approximately 25-fold higher diffusivities of ADMP and Sizzled compared to BMP and Chordin (*Table 1*). To measure the diffusivities of ADMP and Sizzled and test these assumptions, we developed fluorescent ADMP and Sizzled fusion proteins (see Materials and methods). Whereas Sizzled fusion proteins had activity comparable to untagged Sizzled (*Figure 4—figure supplement 1A–C*), ADMP fusions with sfGFP or FLAG tags inserted 2, 5, or 11 amino acids after the Furin cleavage site were much less active than untagged ADMP (data not shown), and could therefore not be used for diffusion measurements. Using FRAP, we measured an effective diffusion coefficient of 9.7 ± 3.2 µm$^2$/s for Sizzled-sfGFP (*Figure 4E*, *Figure 4—figure supplement 1D*). This measurement is consistent with Model 3, but not Model 4, the latter of which requires much higher Sizzled mobility (*Table 1*).

When parameterized with these measured diffusion coefficients and over a ~100-fold range of ADMP diffusion coefficients, Model 3 can form ventral-dorsal gradients over relevant time scales (*Figure 4—figure supplement 1F–J*), but the kinetics of gradient formation do not reflect the measurements of pSmad1/5/9 distribution profiles in *Figure 1A+B*. Moreover, the relatively minor difference between BMP/Chordin and Sizzled diffusivity is not compatible with the 25-fold differential required for Model 4 (*Figure 4—figure supplement 1K–P*).

## Chordin does not regulate BMP protein diffusivity or distribution

Model 5 (Shuttling) postulates that highly diffusive Chordin enhances the mobility of poorly diffusive BMPs (*Ben-Zvi et al., 2008*). In this model, Chordin is secreted dorsally, binds to relatively immobile BMP, and creates a highly mobile BMP/Chordin complex. This complex then diffuses until Chordin is cleaved by a protease (Xlr), rendering BMP immobile again (*Figure 1—figure supplement 1*). To investigate whether Chordin is not only an inhibitor of BMP, but also enhances BMP diffusivity, we increased Chordin levels and measured the effective diffusivity of fluorescent BMP2b. In embryos overexpressing Chordin, we did not observe a significant change in the effective diffusivity of fluorescently tagged BMP2b compared to embryos that did not overexpress Chordin (BMP2b-Dendra2 + Chordin: 2.2 ± 0.2 µm$^2$/s; BMP2b-sfGFP + Chordin: 2.8 ± 0.7 µm$^2$/s; *Figure 4G*). The ability of Chordin to enhance the diffusivity of BMP, a major tenet of Model 5, is therefore not supported by FRAP data.

Model 5 also predicts that Chordin alters the distribution of BMP protein. Over time, the shuttling of BMP by Chordin causes BMP to accumulate away from the Chordin source, resulting in an opposing peak of BMP. Our observation that Chordin does not affect the diffusivity of BMP challenges this view (*Figure 4G*). However, to directly test whether a Chordin source can alter BMP distribution (*Figure 5A+B*), we juxtaposed clones of BMP2b-sfGFP-producing cells with clones of cells secreting untagged Chordin and imaged the formation of the BMP2b-sfGFP gradient over time using light sheet fluorescence microscopy (*Figure 5C+D*, *Videos 9–10*). Model 5 predicts a steeper BMP2b-

sfGFP gradient in the presence of an adjacent Chordin-producing clone compared to a wild type clone (*Figure 5A+B*). Although BMP2b-sfGFP gradients tend to be slightly steeper in the presence of a neighboring Chordin-expressing clone compared to a non-Chordin-expressing clone (*Figure 5D*), this minor change is unlikely to account for the formation of a ventral peak in BMP signaling during the short time (hours) required to complete dorsal-ventral patterning (*Figure 1A+B*). We also failed to observe significant redistribution of BMP in simulations of adjacent BMP and Chordin clones using our measured diffusion coefficients and half-lives (*Figure 5E+F*). This suggests that shuttling of BMP2b by Chordin is not relevant for the early aspects of dorsal-ventral patterning in zebrafish embryos.

## Discussion

The BMP signaling gradient patterns the dorsal-ventral axis during animal development. Five major models can explain how a ventral peak of BMP signaling forms, but the biophysical assumptions underlying these models differ widely (*Table 1*). After experimentally examining these assumptions, our findings lead to four main conclusions. First, Chordin does not play an active role in generating BMP signaling peaks, but only globally inhibits BMP (*Figure 2*). This is consistent with graded source-sink-type models (e.g. Models 1 and 2) and Model 3, but inconsistent with Models 4 and 5 (*Table 1*). Interestingly, BMP signaling in the absence of Chordin is not raised on the extreme dorsal side, indicating that other extracellular inhibitors such as Follistatin or Noggin (*Umulis et al., 2009*) or inhibitors of *bmp* expression (*Koos and Ho, 1999*; *Leung et al., 2003*; *Ramel and Hill, 2013*) that were not included in the tested models might further restrict BMP signaling in these regions. Second, BMP2b and Chordin both diffuse in the extracellular space (*Figure 3F–I*), challenging models involving immobile BMP (Model 2). Third, fluorescently tagged BMP2b and Chordin have similarly high local diffusivities (*Figure 4F*), but on a global scale they move much more slowly through the embryo (*Figure 4E*). These findings rule out Models 2, 3, and 5, but are consistent with Models 1 and 4. Fourth, Chordin does not significantly affect BMP2b diffusion or protein distribution in zebrafish embryos (*Figure 4G*, *Figure 5*), undermining shuttling models in this developmental context. Instead, our data are most consistent with Model 1, the graded source-sink model of BMP/Chordin-mediated dorsal-ventral patterning during early zebrafish development. Our conclusions are also consistent with a recent complementary study (*Zinski et al., 2017*).

Notably, shuttling models (e.g. Model 5) have gained prominence in many developmental contexts including scale-invariant patterning (*Ben-Zvi et al., 2008*; *Barkai and Ben-Zvi, 2009*; *Francois et al., 2009*; *Plouhinec and De Robertis, 2009*; *Ben-Zvi and Barkai, 2010*; *Ben-Zvi et al., 2011a*; *Haskel-Ittah et al., 2012*), but the fundamental tenet, that is, whether putative shuttles such as Chordin change the diffusivity and distribution of signals such as BMP, has not been directly examined. Alternative models that do not invoke Chordin-dependent facilitated BMP diffusion (Model 4) (*Francois et al., 2009*) or that postulate differential protein stability (Model 3) (*Inomata et al., 2013*) can also explain scale-invariant patterning. Our data do not provide strong evidence for shuttling of BMP2b at time scales relevant for dorsal-ventral patterning during early zebrafish embryogenesis: We failed to observe a significant modulation of BMP2b-sfGFP or BMP2b-Dendra2 diffusivity or distribution by Chordin (*Figure 4G*, *Figure 5*). It is, however, possible that other BMPs (e.g. BMP4, BMP7, ADMP) are shuttled by interactions with Chordin and its protease Tolloid/Xlr. Indeed, *tolloid* mutants display a mild patterning defect of the ventral tail fin (*Connors et al., 1999*) that might reflect a requirement for the ventral accumulation of a weakly active, dorsally expressed BMP ligand such as ADMP (*Dickmeis et al., 2001*; *Lele et al., 2001*).

The graded source-sink model (Model 1) that is best supported by our data describes a system in which the graded, ventrally biased distribution of *bmp* mRNA and the dorsally localized *chd* mRNA distribution produce opposing sources of extracellular, diffusing BMP and Chordin protein, which together generate the BMP signaling gradient required for proper dorsal-ventral patterning. Notably, this model fails to take other known dorsal-ventral regulators into account (e.g., ADMP, Sizzled, Follistatin, Noggin). Furthermore, approximately one third of *bmp2b* and *chordin* mutant embryos can be rescued by apparently uniform *bmp* and *chordin* expression, respectively (*Kishimoto et al., 1997*; *Fisher and Halpern, 1999*) (*Figure 3C*), arguing against a strong requirement for concurrent opposing BMP and Chordin sources as long as one component of the system is biased (i.e. ventrally biased *bmp2b* expression with uniform Chordin, or dorsally biased *chordin* expression with uniform

BMP). Thus, further adjustments to the basic Model 1 will be required to fully describe dorsal-ventral patterning.

Although our results support a role for BMP diffusion in dorsal-ventral patterning, the necessity of signal diffusion for developmental patterning has recently been challenged by several studies (*Brankatschk and Dickson, 2006*; *Roy and Kornberg, 2011*; *Alexandre et al., 2014*; *Dominici et al., 2017*; *Varadarajan et al., 2017*). It will be interesting to determine whether BMP diffusion is indeed required for proper patterning using emerging nanobody-mediated diffusion perturbations (*Harmansa et al., 2015*) or optogenetics-based cell-autonomous modulation of signaling range (*Sako et al., 2016*).

# Materials and methods

## Immunostainings

To visualize pSmad1/5/9, wild type TE embryos were dechorionated at the one-cell stage using 1 mg/ml of Pronase (Roche, Cat. No. 11 459 643 001). Dechorionated embryos were incubated at 28°C and fixed at different developmental stages in 4% formaldehyde (Roth) in PBS overnight at 4°C on a shaker. Embryos were then stored in 100% methanol at −20°C for at least 2 hr. All subsequent steps were carried out at room temperature. Embryos were re-hydrated with 70%, 50%, and 30% methanol in PBS for 10 min each. The embryos were then washed eight times with PBST (0.1% Tween) for 15 min and blocked twice with blocking solution (10% fetal bovine serum and 1% DMSO in PBST) for 1 hr, and incubated with 1:100 anti-pSmad1/5/9 antibody (Cell Signaling Technology, Cat. No. 9511) for 4 hr. Embryos were washed with blocking solution for 15 min, washed seven times with PBST, blocked with blocking solution for 1 hr, incubated with 1:500 Alexa 488-coupled goat anti-rabbit secondary antibody (Life Technologies, Cat. No. A11008) for 4 hr, and washed similarly to the procedure after primary antibody application. Embryos were then counterstained with DAPI solution (0.2 µg/ml in PBST) for 1 hr and washed with PBST. Immunostainings were performed using an In situ Pro hybridization robot (Abimed/Intavis).

To analyze pSmad1/5/9 distributions in the absence of Chordin, embryos from one pair of *chordin*[tt250] (*Hammerschmidt et al., 1996*) heterozygous parents were collected, fixed, immunostained with anti-pSmad1/5/9 antibody (Cell Signaling Technology, Cat. No. 13820S) as above, and imaged simultaneously to minimize differences between samples. Embryos were treated as described above, except that progeny from *chordin*[+/-] incrosses were first permeabilized with ice-cold acetone at −20°C for 7 min before the re-hydration step. After imaging and DNA extraction (*Meeker et al., 2007*), progeny from the *chordin*[tt250] heterozygote incross were identified as wild type, heterozygous, or homozygous mutant embryos by PCR amplification using the forward primer 5'-TTCG TTTGGAGGACAACTCG-3' and the reverse primer 5'-AACTCAGCAGCAGAAGTCAATTC-3' with an initial denaturation step of 94°C for 3 min; 39 cycles of 94°C for 30 s, 55°C for 40 s, and 72°C for 30 s; and a final extension at 72°C for 5 min with subsequent digestion with MspI (New England Biolabs, Cat. No. R0106) for 2 hr. The genotyping assay for the *chordin*[tt250] line was designed by the Zebrafish International Resource Center (ZIRC) staff and downloaded from the ZIRC website at http://zebrafish.org.

## Generation of fluorescent BMP2b fusions

All constructs were generated by PCR-based methods (*Horton et al., 1990*), contain the consensus Kozak sequence *gccacc* 5' of the start codon, and were inserted into the EcoRI and XhoI sites of the pCS2(+) vector. To generate BMP2b-sfGFP and BMP2b-Dendra2, sequences encoding sfGFP or Dendra2 flanked by LGDPPVAT linkers were inserted two amino acids downstream of the BMP2b Furin cleavage site. Sequences encoding the FLAG tag DYKDDDDK were inserted between the first linker and sfGFP or Dendra2 to generate BMP2b-sfGFP-FLAG and BMP2b-Dendra2-FLAG. To generate BMP2b-FLAG, the FLAG tag was inserted between two LGDPPVAT linkers two amino acids downstream of the BMP2b Furin cleavage site.

## Generation of fluorescent Chordin fusions

All constructs were generated by PCR-based methods (*Horton et al., 1990*) and contain the consensus Kozak sequence *gccacc* 5' of the start codon. Chordin was inserted into the ClaI site of pCS2(+).

All other Chordin-containing constructs were inserted into the EcoRI and XbaI sites of the pCS2(+) vector. To generate Chordin-sfGFP and BMP2b-Dendra2, sequences encoding sfGFP or Dendra2 flanked by LGDPPVAT linkers were inserted immediately 5' of the Tolloid cleavage site 2. To generate Chordin-FLAG, sequences encoding the FLAG tag DYKDDDDK were inserted immediately 5' of the Tolloid cleavage site 2 without additional linkers. To generate Chordin-sfGFP-FLAG and Chordin-Dendra2-FLAG, sequences encoding the FLAG tag were inserted between the first linker and sfGFP or Dendra2 of Chordin-sfGFP and Chordin-Dendra2 constructs.

## Generation of fluorescent Sizzled fusions

All Sizzled constructs were generated by PCR-based methods (*Horton et al., 1990*), contain the consensus Kozak sequence *gccacc* 5' of the start codon, and were inserted into the EcoRI and XbaI sites of the pCS2(+) vector. To generate Sizzled-sfGFP, sequences encoding sfGFP with an N-terminal LGLG linker were fused to the C-terminus of Sizzled. Sequences encoding the FLAG tag DYKDDDDK were inserted between the LGLG linker and sfGFP to generate Sizzled-sfGFP-FLAG. To generate Sizzled-FLAG, the FLAG tag was fused to the C-terminus of Sizzled separated by an LGLG linker.

## mRNA *in vitro* synthesis

mRNA was generated using SP6 mMessage mMachine kits (Thermo Fisher) after vector linearization with NotI-HF (New England Biolabs, Cat. No. R3189). mRNA was purified using LiCl precipitation or Qiagen RNeasy kits following the manufacturers' instructions.

## Phenotypic analysis

Scoring of ventralization and dorsalization was executed as previously described (*Mullins et al., 1996*; *Kishimoto et al., 1997*). Embryos were injected at the one- to two-cell stage with equimolar amounts of *BMP2b* (1 pg), *BMP2b-sfGFP* (1.49 pg), and *BMP2b-Dendra2* (1.47 pg) mRNA to assess ventralizing activity. At 1 day post-fertilization, BMP2b-injected embryos were classified as weakly ventralized (V1) to strongly ventralized (V4). V1 embryos have reduced eyes but a prominent head. V2 embryos have no eyes, reduction of the head, and expansion of posterior structures such as somites and tail. V3 embryos completely lack head structures and exhibit a further expanded tail and enlarged blood islands. Finally, V4 embryos lack most structures except for a short, protruding, and expanded tail.

To assess dorsalizing activity of the Chordin constructs, embryos were injected with equimolar amounts of *Chordin* (30 pg), *Chordin-sfGFP* (37 pg), *Chordin-Dendra2* (37 pg), and *Chordin-FLAG* mRNA (30 pg). Embryos were scored at 1 day post-fertilization and classified as weakly dorsalized (C1) to strongly dorsalized (C5) (*Kishimoto et al., 1997*). C1 embryos lack the ventral tail fin. C2 embryos have a further loss of ventral structures, such as the ventral tail vein, and a bent tail. C3 embryos exhibit a tail that is shortened and twisted. C4 embryos have observable head structures and develop eyes with twisting of the posterior structures above the yolk. C5 embryos are fully dorsalized and frequently lyse (*Mullins et al., 1996*; *Kishimoto et al., 1997*).

## Rescue of *BMP2b* (*swr$^{-/-}$*) mutants

Injection of *BMP2b* mRNA can rescue *BMP2b* mutants (*Kishimoto et al., 1997*). To investigate whether tagged BMP2b constructs can rescue *swr$^{tc300-/-}$* mutants (*Mullins et al., 1996*), the rescuing amount of *BMP2b* mRNA was first determined (1.8 pg), and equimolar amounts of mRNA encoding fluorescent fusion constructs were subsequently injected into the progeny of heterozygous *swr$^{+/-}$* mutant incrosses. Embryos with wild type morphology at 24 hpf were anesthetized and mounted in 2% methylcellulose for imaging with an AxioZoom V16 (ZEISS) microscope at 30–33 hpf. To genotype embryos following DNA extraction (*Meeker et al., 2007*), PCR was performed to amplify a BMP2b fragment with the forward primer 5'-AAAAGCCGAGGAGAAAGCAC-3' and the reverse primer 5'-AGTCCTTCATTGGGGAGATTGTTC-3', and the following thermocycling parameters: An initial denaturation step of 94°C for 3 min; 39 cycles of 94°C for 30 s, 58°C for 40 s, and 72°C for 40 s; and a final extension at 72°C for 5 min. PCR amplicons were subsequently digested with HaeIII (New England Biolabs, Cat. No. R0108) at 37°C for 2 hr. The genotyping assay for the

*swr^tc300* line was designed by the Zebrafish International Resource Center (ZIRC) staff and downloaded from the ZIRC website at http://zebrafish.org.

## Preparation of extracellularly enriched fractions for western blotting

Extracellularly enriched and cellular fractions from manually deyolked embryos between sphere and dome stage were obtained as described previously (*Müller et al., 2012*). mRNAs encoding FLAG-tagged constructs were injected at the one- or two-cell stage at equimolar amounts (*BMP2b-FLAG*: 444 pg, *BMP2b-sfGFP-FLAG*: 638 pg, *BMP2b-Dendra2-FLAG*: 630 pg; and *Chordin-FLAG*: 500 pg, *Chordin-sfGFP-FLAG*: 620 pg, *Chordin-Dendra2-FLAG*: 615 pg). For protein samples with BMP2b constructs, fractions from approximately 19 embryos were loaded and resolved by SDS-PAGE using 12% polyacrylamide gels. For protein samples with Chordin constructs, fractions from approximately 17–18 embryos were loaded and resolved in 8% polyacrylamide gels. Proteins were subsequently transferred onto PVDF membranes using a Trans-Blot Turbo Transfer System (Bio-Rad, Cat. No. 170–4272). Membranes were blocked with 5% non-fat milk (Roth, Cat. No. T145.2) in PBST (0.1% Tween) and incubated with anti-FLAG antibody (Sigma, Cat. No. F3165) at a concentration of 1:2000 in non-fat milk in PBST at 4°C overnight. HRP-coupled donkey anti-mouse secondary antibody (Jackson ImmunoResearch, Cat. No. 715-035-150) was used at concentration of 1:25,000 for 3 hr at room temperature. Chemiluminescence was detected using SuperSignal West Dura Extended Duration Substrate (Thermo Fisher, Cat. No. 34075) and imaged with a chemiluminescence imaging system (Fusion Solo, Vilber Lourmat).

## Transplantations

To generate clonal sources secreting BMP2b-sfGFP, Chordin-sfGFP, and untagged Chordin (*Figures 3* and *5*), approximately 50–75 cells were transplanted from sphere stage wild type TE donor embryos expressing these constructs into uninjected, sphere stage sibling hosts (similar to [*Müller et al., 2012*]). Transplantations were carried out in 1 x Ringer's buffer. Cells were explanted from donors, extruded briefly into the buffer to wash away cellular debris and extracellular fluorescent protein, and then transplanted into host embryos.

Donor embryos were dechorionated with 1 mg/ml Pronase (Roche, Cat. No. 11 459 643 001) and injected with 1–2 nl injection mix at the one-cell stage. Sibling host embryos were dechorionated together with donors at the one-cell stage, and all embryos were incubated at 28°C until transplantation. Unfertilized or injured embryos were discarded.

For single (*Figure 3*) and double (*Figure 5*) transplantation experiments, BMP2b-sfGFP and Chordin-sfGFP donors were injected with 500 pg mRNA (*Figure 3—figure supplement 1F–H*).

For double transplantation experiments (*Figure 5*), embryos received one transplantation from a donor expressing BMP2b-sfGFP and a second transplantation from a donor injected at the one-cell stage with either 50 pg Alexa 546-coupled dextran (10 kDa, Molecular Probes, Cat. No. D22911) or 1000 pg Chordin mRNA + 50 pg Alexa 546-coupled dextran. Alexa 546-coupled dextran was used to mark the location of the second clone.

2–10 min post-transplantation, embryos were mounted in 1% low-melting NuSieve GTG agarose (Lonza, Cat. No. 50080) dissolved in embryo medium (250 mg/l Instant Ocean salt dissolved in reverse osmosis water). Embryos were immersed in 40°C molten low melting point agarose, pulled into 1.5 mm glass capillary tubes (ZEISS), and positioned with the animal pole perpendicular to the capillary using a metal probe. Agarose tubes were then suspended in embryo medium, and imaged at room temperature using a ZEISS Lightsheet Z.1 microscope (see *Light sheet microscopy* section for further imaging details).

## Light sheet microscopy

Fluorescence images in *Figures 1*, *2*, *3* and *5*, and *Figure 3—figure supplement 1* were obtained using a Lightsheet Z.1 microscope (ZEISS). For fixed, immunostained embryos, samples were mounted into a glass capillary sample holder in 1% low-melting NuSieve GTG agarose (Lonza, Cat. No. 50080) in embryo medium with 0.2 μm dark red fluorescent FluoSpheres (Life Technologies, Cat. No. F8807) diluted 1:200,000 from a 2% solids stock. Embryos were imaged at 0°, 45°, 180° and 225° angles (*Schmid et al., 2013*) using identical imaging conditions. For 3D reconstruction, an interactive bead-based registration algorithm was used to determine the threshold that most accurately

selects the beads (*Preibisch et al., 2010*). Reconstructed images were converted to 8-bit format using ImageJ, and Imaris software (Bitplane) was used for 3D data visualization and video generation. The videos were cropped using Avidemux 2.6.

To visualize the entire embryo in a single image, reconstructed images were first converted to 16-bit files using ImageJ, and equirectangular 2D map projections were then generated (*Schmid et al., 2013*). The 2D maps were re-aligned into Hammer-Aitoff projections using Hugin panorama photo stitcher software (http://hugin.sourceforge.net) to orient the peak of pSmad1/5/9 intensity to the ventral pole (left in *Figure 1* panels) and the trough of pSmad1/5/9 intensity to the dorsal pole (right in *Figure 1* panels). For gradient quantifications in *Figure 1A+B* and *Figure 2F–H*, the embryo proper was masked using manual polygon selections in Fiji (*Schindelin et al., 2012*) in order to exclude signal from the yolk syncytial layer and yolk. The 'Plot Profile' function in Fiji was then applied to the entire masked image to determine ventral-to-dorsal gradients. The background signal of immunostained embryos was determined by finding the lowest value in the profiles of sphere stage embryos (*Figure 1A+B*) and the lowest value in the profiles of *chordin*$^{-/-}$ embryos (*Figure 2F +G*), respectively. These background values were subtracted from the data sets, and the profiles were normalized to the highest value in each data series. The mean and standard error of the normalized data sets was then calculated piece-wise for every point along the ventral-to-dorsal profile.

For transplantation experiments in *Figures 3* and *5*, imaging began 5 to 20 min post-transplantation and continued for approximately 1 hr (see *Transplantation* section for further details). The following imaging conditions were used:

- W Plan-Apochromat 20 x objective, 0.5 x zoom
- dual side light sheets
- 488 nm laser (100 mW) at 6% power (for sfGFP-containing constructs)
- 561 nm laser (20 mW) at 5% power (for double transplantations only; to detect Alexa 546 signal)
- separate exposure to 488/561 nm lasers (in double transplantation experiments only) to avoid cross-talk
- exposure time: 250 ms
- average light sheet thickness: 6.4 µm
- 3 µm intervals between z-slices; 60 slices per embryo ($\approx$180 µm total)
- 5 min intervals between imaging

Gradients were quantified using maximum intensity projections of 15 z-slices similar to the approach in (*Müller et al., 2012*). A rectangular region of interest abutting the clone with a fixed height of 86.34 µm (corresponding to 189 pixels) and varying widths depending on embryo length was drawn in Fiji (*Schindelin et al., 2012*), and the average intensity in 0.457 µm strips was calculated from the maximum intensity projections. Background intensity resulting from autofluorescence was measured similarly in uninjected embryos (for single transplantation experiments, n = 4) or in uninjected embryos transplanted with a clone of cells containing Alexa 546-coupled dextran (for double transplantation experiments, n = 2). A single value for background subtraction was determined by calculating the average of the intensity profile values. After subtracting the background value from the experimental intensity profiles, the data was normalized to the value closest to the clonal source boundary. This approach allows for the comparison of the relative gradient range, which is independent of constant production rates. We assume constant production rates over the relatively short time scales of observation ($\approx$80 min).

Embryos with low signal-to-noise ratios were excluded from analysis.

## Fluorescence decrease after photoconversion (FDAP) experiments

FDAP experiments were carried out as described in (*Müller et al., 2012*; *Rogers et al., 2015*). Embryos were injected at the one-cell stage with either 60 pg *BMP2b-Dendra2* mRNA + 0.5 ng Alexa 488-dextran (3 kDa, Molecular Probes) or 150 pg *Chordin-Dendra2* mRNA + 0.5 ng Alexa 488-dextran. To assess background fluorescence, embryos were injected with 0.5 ng Alexa 488-dextran only. Embryos were mounted in 1% low melting point agarose in glass-bottom Petri dishes (MatTek Corporation) covered with embryo medium to hydrate the agarose during imaging.

FDAP experiments were performed using an LSM 780 (ZEISS) confocal microscope. Pre-conversion and post-conversion images were acquired using an LD C-Apochromat 40x/1.1 NA water

immersion objective. A single pre-photoconversion image was first acquired for each sample followed by photoconversion and multiposition time-lapse imaging with 10 min intervals for approximately 300 min. For photoconversion, embryos were illuminated with a Sola SE II LED lamp at 100% power for 30 s through a C-Apochromat 10x/0.45 NA objective and an AHF F36-500 UV filter cube. For both pre- and post-conversion images, Alexa 488 was excited using a 488 nm Argon laser, and a DPSS 561 nm laser was used to excite photoconverted Dendra2. The emission signal between 494–576 nm (Alexa 488) and 578–696 nm (photoconverted Dendra2) was collected using a 32 channel GaAsP QUASAR detector array. Embryos that produced only low levels of photoconverted Dendra2 signal or whose position shifted significantly over time as well as embryos with non-uniform signal distribution or embryos that died were excluded from analysis. Sample numbers: n = 22 for BMP2b-Dendra2 (with n = 17 background embryos); n = 6 for Chordin-Dendra2 (with n = 1 background embryo).

All experiments were analyzed using PyFDAP (*Bläßle and Müller, 2015*; *Rogers et al., 2015*), version 1.1.2. PyFDAP extracts the extracellular and intracellular photoconverted Dendra2 signal by masking the Alexa 488 signal, and fits the resulting average intensities with a linear decay model. The ordinary differential equation describing linear protein decay is given by

$$\frac{dc}{dt} = -k_1 c$$

where $c$ is the concentration of the protein and $k_1$ is its clearance rate constant. We assume that Dendra2 signal is directly proportional to the protein concentration. The analytical solution of this equation is given by

$$c(t) = c_0 e^{-k_1 t} + y_0$$

where $c_0 + y_0$ is the protein's concentration at $t = 0$, and $y_0$ is the protein's concentration at $t = \infty$. The half-life $\tau$ of the protein can then be calculated as

$$\tau = \ln(2)/k$$

PyFDAP estimates a lower bound for $y_0$ by computing the maximum relative effect of photobleaching $F_{i,r}$. For each background data set, the strongest influence of photobleaching was computed by taking the minimum over all differences of background intensity $B_{j,r}$ and background noise $N_i$, and the difference between pre-conversion background intensity $B_{\text{pre},i,r}$ and noise level. Here, $r$ denotes the region under consideration, i.e. extracellular, intracellular, or the entire imaging slice; $i$ indicates the $i$th data set, and $j$ counts the background data sets. The average over all $b$ background data sets was then taken to arrive at the mean effect of photobleaching. The factor

$$F_{i,r} = \frac{1}{b}\sum_{j=1}^{b} \min_t \left( \frac{B_{j,r}(t) - N_i}{B_{\text{pre}_{j,r}} - N_i} \right)$$

was used to scale the pre-conversion intensity of the FDAP data set according to

$$y_{0_{i,r}} \geq F_{i,r}\left(I_{\text{pre}_{i,r}} - N_i\right) + N_i$$

This lower bound was then used to constrain a Nelder-Mead simplex algorithm when minimizing

$$SSD = \sum_n \left(\bar{I}(t_n) - c(t_n)\right)^2$$

## Fluorescence recovery after photobleaching (FRAP) experiments

FRAP experiments and data analysis were carried out as previously described (*Müller et al., 2012*; *Müller et al., 2013*) using an LSM 780 NLO confocal microscope (ZEISS) and an LD LCI Plan-Apochromat 25x water immersion objective. Embryos were injected at the one-cell stage with 30 pg of mRNA encoding BMP2b-sfGFP, 60 pg of mRNA encoding BMP2b-Dendra2, 60 pg of mRNA encoding Chordin-sfGFP, 120 pg of mRNA encoding Chordin-Dendra2, or 30 pg of mRNA encoding Sizzled-sfGFP. To analyze the effect of Chordin on BMP2b diffusion, embryos were injected at the one-cell stage with 30 pg of mRNA encoding BMP2b-sfGFP plus 60 or 200 pg of mRNA encoding

Chordin, or 60 pg of mRNA encoding BMP2b-Dendra2 plus 200 pg of mRNA encoding Chordin. Embryos were mounted in 1% low-melting point agarose in glass-bottom Petri dishes (MatTek Corporation) covered with embryo medium to hydrate the agarose during imaging. Embryos with low or non-uniform fluorescence and embryos that died or whose position shifted significantly over time were excluded from analysis.

For FRAP data analysis, the fits of a model with uniform production, diffusion, and clearance were constrained with the clearance rate constants of BMP2b-Dendra2 and Chordin-Dendra2 fusions measured by FDAP in the present study (BMP2b-Dendra2: $k_1$ = 8.9 × $10^{-5}$/s; Chordin-Dendra2: $k_1$ = 9.6 × $10^{-5}$/s). Sizzled-sfGFP fits were constrained with the clearance constant measured for BMP2b-Dendra2 assuming similar protein stability. As shown previously, the estimation of diffusion coefficients does not sensitively depend on the values of clearance rate constants if the time scales of observation (here: 50 min) and protein stability (here: approximately 120 min) are similar (*Müller et al., 2012*).

## Fluorescence correlation spectroscopy (FCS) experiments

The FCS experiments were done using an LD C-Apochromat 40x/1.1 NA water immersion objective on an LSM 780 NLO confocal microscope (ZEISS). Embryos were injected at the one-cell stage with 30 pg of mRNA encoding BMP2b-sfGFP or 60 pg of mRNA encoding Chordin-sfGFP. Embryos were mounted in 1% low-melting point agarose in glass-bottom Petri dishes (MatTek Corporation) and covered with embryo medium to hydrate the agarose during imaging. The fluorophores (sfGFP, Alexa 488) were excited using an Argon 488 nm laser, and the emission light between 494 and 542 nm was collected using a 32-channel GaAsP QUASAR detector array. Before each FCS experiment, the pinhole was aligned and set to 1 Airy unit, and the instrument was calibrated using a solution of 40 nM Alexa 488 dye (Thermo Fisher) in water. For each FCS sample, fluorescence fluctuations were measured for 10 s with 10 repeats, and any irregularities in the 100 s count trace resulting from cellular movements were excluded from analysis.

Auto-correlation curves for Alexa 488 were freely fitted to determine the structural parameter as well as the diffusion time, the triplet state fraction, and the triplet state relaxation time of Alexa 488 for every experiment. The auto-correlation curves for BMP2b-sfGFP and Chordin-sfGFP were fitted with a fixed structural parameter, fixed triplet state fraction, and fixed triplet relaxation time determined from the Alexa 488 calibration measurements. The curves were fitted using ZEISS ZEN Pro software with a one-component 'free diffusion with triplet state correction' model. The first $10^{-6}$ seconds lag time for the correlation curve was excluded in the fitting (*Yu et al., 2009*; *Müller et al., 2013*). The diffusion coefficient was then calculated by comparing the diffusion time of BMP2b-sfGFP and Chordin-sfGFP with Alexa 488 (reference diffusion coefficient: 435 μm²/s [*Petrásek and Schwille, 2008*]).

Since the values of the triplet state fraction and the triplet state relaxation time of sfGFP are unknown and not necessarily identical to those of Alexa 488, we also freely fitted the autocorrelation curves for BMP2b-sfGFP and Chordin-sfGFP with the experimentally measured structural parameter as the only constraint, and determined free diffusion coefficients of $D$ = 35 ± 2 μm²/s for BMP2b-sfGFP (n = 17 measurements from 4 embryos) and $D$ = 50 ± 3 μm²/s for Chordin-sfGFP (n = 19 measurements from 5 embryos), within a deviation of approximately 20–30% compared to the diffusion coefficients determined by constraining the fits with a fixed structural parameter, fixed triplet state fraction, and fixed triplet relaxation time ($D$ = 46 ± 1 μm²/s for BMP2b-sfGFP, and $D$ = 59 ± 2 μm²/s for Chordin-sfGFP; values reported in *Figure 4*). The similar diffusion coefficients determined by differently constrained fits indicate that the diffusion time measured in our experiments does not sensitively depend on the values of the triplet state fraction and triplet state relaxation time.

## Mathematical modeling of BMP2b-sfGFP and Chordin-sfGFP gradient formation

The geometry of the zebrafish blastoderm was approximated by the complement of two spheres with a columnar subdomain placed off-center to represent the signal source region with the same parameters as described in *Müller et al. (2012)*. Gradient formation was simulated with the source-diffusion-sink model

$$\frac{\partial c}{\partial t} = D\nabla^2 c - k_1 c + \delta_s k_2$$

with

$$\delta_s = \begin{cases} 1 & \text{in the source} \\ 0 & \text{otherwise} \end{cases}$$

For *Figure 3—figure supplement 2*, the experimental data were fitted with solutions from a 50 × 50 parameter grid spanning all possible combinations of 50 diffusion coefficients (logarithmically spaced from 0.1 µm$^2$/s to 50 µm$^2$/s) and 50 clearance rate constants (logarithmically spaced from 1 × 10$^{-5}$/s to 5 × 10$^{-4}$).

## Simulations of previous models

The finite element method was used for all numerical simulations. All geometries are one-dimensional representations of embryos. The solution at each time step in the discretized geometries was determined using a sparse LU factorization algorithm (UMFPACK), and the time stepping was computed using a backward Euler step method (Comsol Multiphysics). Simulations in *Figure 1C–E,G* (Models 1, 2, 3, and 5) were executed for a total of 10080 s (i.e., for approximately 3 hr from sphere to shield stage during zebrafish embryogenesis [*Kimmel et al., 1995*]) and read out every 2520 s (i. e., approximately every 42 min at relevant zebrafish stages). The simulation in *Figure 1F* (Model 4) was executed for a total of 20 time steps near steady state and read out at every fifth time step.

The following model descriptions comprise the complete wild type systems. For simulations of *chordin* mutants, the Chordin flux was set to 0 (Models 1, 2, 3, and 5), or the Chordin-dependent terms were removed from the equations and the initial concentration of Chordin was set to 0 (Model 4). To focus on the role of Chordin in regulating BMP signaling and distribution, we did not include other negative regulators of BMP such as Noggin and Follistatin (*Umulis et al., 2009*). For the interpretation of the simulations, we assume that the distribution of free BMPs is correlated with BMP signaling and the distribution of pSmad1/5/9.

To facilitate comparison of the models, the distribution profiles of free BMP are shown as a function of relative embryo length, and the solutions were normalized to the ventral-most free BMP concentration at shield stage (i.e., at $t = 7560$ s for Models 1, 2, 3, and 5, and at $t = 15$ for Model 4) in wild type simulations.

## Model 1: Graded source-sink (mobile BMP)

In the graded source-sink model, the BMP source $\rho_{BMP}(x)$ was modeled after the known distribution of *bmp2b* mRNA between sphere stage and 30% epiboly (*Ramel and Hill, 2013*). The model does not include autoregulation of BMP production since positive feedback only appears to be important for later stages of development (*Ramel and Hill, 2013*; *Zinski et al., 2017*). Chordin binds BMP irreversibly and acts as a sink. The model was simulated using the following equations:

$$\frac{\partial[\text{BMP}]}{\partial t} = D_{\text{BMP}}\nabla^2[\text{BMP}] - \kappa[\text{Chd}][\text{BMP}] - \lambda_{\text{BMP}}[\text{BMP}] + \rho_{\text{BMP}}(x)$$

$$\frac{\partial[\text{Chd}]}{\partial t} = D_{\text{Chd}}\nabla^2[\text{Chd}] - \kappa[\text{Chd}][\text{BMP}] - \lambda_{\text{Chd}}[\text{Chd}]$$

$$\frac{\partial[\text{ChdBMP}]}{\partial t} = D_{\text{ChdBMP}}\nabla^2[\text{ChdBMP}] + \kappa[\text{Chd}][\text{BMP}] - \lambda_{\text{Chd}}[\text{ChdBMP}]$$

### Embryo geometry and boundary conditions
Embryo length: $300 \times 10^{-6}$ m (300 µm, the typical length of the zebrafish blastoderm)
Constant Chordin flux from the dorsal boundary: $5 \times 10^{-14}$ mol/(m$^2$•s)
No-flux boundary condition for all other species on both ventral and dorsal boundaries

### Parameter values
$D_{BMP}$ = 2 µm$^2$/s (measured in the present study)
$D_{Chd}$ = 7 µm$^2$/s (measured in the present study)
$D_{ChdBMP}$ = 7 µm$^2$/s

$\lambda_{BMP} = 8.9 \times 10^{-5}$/s (measured in the present study)
$\lambda_{Chd} = 9.6 \times 10^{-5}$/s (measured in the present study)
$\kappa = 400 \times 10^3$ m$^3$/(mol·s)
$\rho_{BMP}(x) = 0.57 \times 10^{-9} \times e^{-5000x}$ mol/m$^3$ (accounting for the inhomogeneous ventrally peaking distribution of *bmp2b* mRNA in zebrafish embryos)

### Initial conditions
BMP initial concentration: $2.85 \times 10^{-8}$ mol/m$^3$ everywhere (one-twentieth of the concentration used for *Xenopus* frogs in [*Inomata et al., 2013*])
Chordin initial concentration: 0 mol/m$^3$ everywhere
Chordin-BMP complex initial concentration: 0 mol/m$^3$ everywhere

## Model 2: Graded source-sink (immobile BMP)
As for Model 1, the graded source-sink model (immobile BMP) was modeled without autoregulation of BMP production since positive feedback only appears to be important for later stages of development (*Ramel and Hill, 2013*; *Zinski et al., 2017*). Here $\kappa$, which reflects the binding between Chordin and BMP, is smaller than in Model 1 to obtain a realistic-free BMP distribution; using the same value for $\kappa$ as in Model 1 creates an unrealistically steep free BMP gradient. The model was simulated using the following equations:

$$\frac{\partial[\text{BMP}]}{\partial t} = -\kappa[\text{Chd}][\text{BMP}] - \lambda_{\text{BMP}}[\text{BMP}] + \rho_{\text{BMP}}(x)$$

$$\frac{\partial[\text{Chd}]}{\partial t} = D_{\text{Chd}}\nabla^2[\text{Chd}] - \kappa[\text{Chd}][\text{BMP}] - \lambda_{\text{Chd}}[\text{Chd}]$$

$$\frac{\partial[\text{ChdBMP}]}{\partial t} = D_{\text{ChdBMP}}\nabla^2[\text{ChdBMP}] + \kappa[\text{Chd}][\text{BMP}] - \lambda_{\text{Chd}}[\text{ChdBMP}]$$

### Embryo geometry and boundary conditions
Embryo length: $300 \times 10^{-6}$ m (300 µm, the typical length of a zebrafish blastoderm)
Constant Chordin flux from the dorsal boundary: $5 \times 10^{-14}$ mol/(m$^2$·s)
No-flux boundary condition for all other species on both ventral and dorsal boundaries

### Parameter values
$D_{Chd}$ = 7 µm$^2$/s (measured in the present study)
$D_{ChdBMP}$ = 7 µm$^2$/s
$\lambda_{BMP} = 8.9 \times 10^{-5}$/s (measured in the present study)
$\lambda_{Chd} = 9.6 \times 10^{-5}$/s (measured in the present study)
$\kappa = 4 \times 10^3$ m$^3$/(mol·s)
$\rho_{BMP}(x) = 0.57 \times 10^{-9} \times e^{-5000x}$ mol/m$^3$ (accounting for the inhomogenous ventrally peaking distribution of *bmp2b* mRNA in zebrafish embryos)

### Initial conditions
BMP initial concentration: $2.85 \times 10^{-8}$ mol/m$^3$ everywhere (one-twentieth of the concentration used for *Xenopus* frogs in [*Inomata et al., 2013*]).
Chordin initial concentration: 0 mol/m$^3$ everywhere
Chordin-BMP complex initial concentration: 0 mol/m$^3$ everywhere

## Model 3: Long-range accumulation and feedback
The model was developed for frog embryogenesis. For the simulations in the present study the equations, geometry, initial conditions, and parameters used were exactly as described in (*Inomata et al., 2013*):

$$\frac{\partial [\text{BMP}]}{\partial t} = D\nabla^2[\text{BMP}] + \frac{v_{\text{BMP}}([\text{ADMP}] + [\text{BMP}])^{10}}{k_{\text{BMP}}^{10} + ([\text{ADMP}] + [\text{BMP}])^{10}} - \lambda_{\text{BMP}}[\text{BMP}]$$

$$+ \frac{\lambda_{\text{Chd}}[\text{ChdBMP}]}{1 + \frac{[\text{Szl}]}{ki} + \frac{[\text{Chd}] + [\text{ChdBMP}] + [\text{ChdADMP}]}{km}} - k[\text{Chd}][\text{BMP}]$$

$$\frac{\partial [\text{Chd}]}{\partial t} = D\nabla^2[\text{Chd}] + \frac{v_{\text{Chd}}k_{\text{Chd}}^{10}}{k_{\text{Chd}}^{10} + ([\text{ADMP}] + [\text{BMP}])^{10}} - \frac{\lambda_{\text{Chd}}[\text{Chd}]}{1 + \frac{[\text{Szl}]}{ki} + \frac{[\text{Chd}] + [\text{ChdBMP}] + [\text{ChdADMP}]}{km}}$$

$$- k[\text{Chd}][\text{BMP}] - k[\text{Chd}][\text{ADMP}]$$

$$\frac{\partial [\text{ADMP}]}{\partial t} = D\nabla^2[\text{ADMP}] + \frac{v_{\text{ADMP}}k_{\text{ADMP}}^{10}}{k_{\text{ADMP}}^{10} + ([\text{ADMP}] + [\text{BMP}])^{10}} - \lambda_{\text{BMP}}[\text{ADMP}]$$

$$+ \frac{\lambda_{\text{Chd}}[\text{ChdADMP}]}{1 + \frac{[\text{Szl}]}{ki} + \frac{[\text{Chd}] + [\text{ChdBMP}] + [\text{ChdADMP}]}{km}} - k[\text{Chd}][\text{ADMP}]$$

$$\frac{\partial [\text{Szl}]}{\partial t} = D\nabla^2[\text{Szl}] + \frac{v_{\text{Szl}}([\text{ADMP}] + [\text{BMP}])^{20}}{k_{\text{Szl}}^{20} + ([\text{ADMP}] + [\text{BMP}])^{20}} - \lambda_{\text{Szl}}[\text{Szl}]$$

$$\frac{\partial [\text{ChdBMP}]}{\partial t} = D\nabla^2[\text{ChdBMP}] - \frac{\lambda_{\text{Chd}}[\text{ChdBMP}]}{1 + \frac{[\text{Szl}]}{ki} + \frac{[\text{Chd}] + [\text{ChdBMP}] + [\text{ChdADMP}]}{km}} + k[\text{Chd}][\text{BMP}]$$

$$\frac{\partial [\text{ChdADMP}]}{\partial t} = D\nabla^2[\text{ChdADMP}] - \frac{\lambda_{\text{Chd}}[\text{ChdADMP}]}{1 + \frac{[\text{Szl}]}{ki} + \frac{[\text{Chd}] + [\text{ChdBMP}] + [\text{ChdADMP}]}{km}} + k[\text{Chd}][\text{ADMP}]$$

## Embryo geometry and boundary conditions

Embryo length: $1000 \times 10^{-6}$ m (1000 μm, the typical length of a frog embryo)
Constant Chordin flux from the dorsal boundary: $4.8 \times 10^{-12}$ mol/(m²•s)
No-flux boundary condition for all other species on both ventral and dorsal boundaries

## Parameter values

$km = 25 \times 10^{-6}$ mol/m³
$ki = 25 \times 10^{-6}$ mol/m³
$v_{Chd} = 5 \times 10^{-10}$ mol/(m³•s)
$k_{Chd} = 7 \times 10^{-8}$ mol/m³
$v_{BMP} = 1.4 \times 10^{-10}$ mol/(m³•s)
$k_{BMP} = 3.5 \times 10^{-7}$ mol/m³
$v_{Szl} = 100 \times 10^{-6}$ mol/(m³•s)
$k_{Szl} = 1 \times 10^{-6}$ mol/m³
$v_{ADMP} = 3.2 \times 10^{-9}$ mol/(m³•s)
$k_{ADMP} = 3 \times 10^{-8}$ mol/m³
$\lambda_{Chd} = 1 \times 10^{-3}$/s
$\lambda_{BMP} = 2 \times 10^{-4}$/s
$\lambda_{Szl} = 3.8 \times 10^{-5}$/s
$D = 15$ μm²/s
$k = 280$ m³/(mol•s)

## Initial conditions

BMP initial concentration: $0.57 \times 10^{-6} \times e^{-1000x}$ mol/m³ throughout the embryo (the amplitude of this distribution is the same as in [*Inomata et al., 2013*], but the initial BMP profile was modeled as a gradient instead of uniform)
Chordin initial concentration: 0 mol/m³ everywhere

ADMP initial concentration: 0 mol/m$^3$ everywhere
Sizzled initial concentration: 0 mol/m$^3$ everywhere
Chordin-BMP complex initial concentration: 0 mol/m$^3$ everywhere
Chordin-AMP complex initial concentration: 0 mol/m$^3$ everywhere

For the simulations in *Figure 4—figure supplement 1E–J*, all parameters were identical to the parameter values listed above except for $D$(BMP) = 3 μm$^2$/s, $D$(Chd) = 6 μm$^2$/s, $D$(ChdADMP) = 10 μm$^2$/s, and $D$(ChdBMP) = 10 μm$^2$/s. $D$(Sizzled) was set to 150 μm$^2$/s in *Figure 4—figure supplement 1E*, and to 10 μm$^2$/s in *Figure 4—figure supplement 1F–J*. $D$(ADMP) was varied from 0.1 μm$^2$/s to 150 μm$^2$/s as indicated in *Figure 4—figure supplement 1E–J*.

## Model 4: Self-regulating reaction-diffusion system

The non-dimensional model, geometry, initial conditions, and parameters used for the simulations were similar to the ones described in [*Francois et al., 2009*]:

$$\frac{\partial[\text{BMP}]}{\partial t} = D_{\text{BMP}}\nabla^2[\text{BMP}] + \frac{[\text{BMP}]^2}{(1+[\text{Chd}])[\text{Szl}]} - \mu_{\text{BMP}}[\text{BMP}] + \rho_{\text{BMP}}$$

$$\frac{\partial[\text{Chd}]}{\partial t} = D_{\text{Chd}}\nabla^2[\text{Chd}] + \frac{[\text{Chd}]^2}{[\text{ADMP}]} - \mu_{\text{Chd}}[\text{Chd}] + \rho_{\text{Chd}}$$

$$\frac{\partial[\text{ADMP}]}{\partial t} = D_{\text{ADMP}}\nabla^2[\text{ADMP}] + [\text{Chd}]^2 - \mu_{\text{ADMP}}[\text{ADMP}]$$

$$\frac{\partial[\text{Szl}]}{\partial t} = D_{\text{Szl}}\nabla^2[\text{Szl}] + [\text{BMP}]^2 - \mu_{\text{Szl}}[\text{Szl}]$$

## Embryo geometry and boundary conditions

Embryo length: 25
No-flux boundary conditions on the ventral and dorsal boundaries

## Parameter values

$D_{Chd} = D_{BMP} = 6$
$\mu_{Chd} = \mu_{BMP} = 1.2$
$\rho_{Chd} = \rho_{BMP} = 0.1$
$\mu_{ADMP} = \mu_{Szl} = 1.5$
$D_{ADMP} = D_{Szl} = 150$

## Initial conditions

BMP initial concentration: $\rho_{BMP} = e^{-0.1x}$
Chordin initial concentration of 1 from position 0 to 24 and Chordin initial concentration of 10 from 24 to 25 (i.e., the dorsal organizer) in the simulated embryo
ADMP initial concentration: 1 everywhere
Sizzled initial concentration: 1 everywhere

For the simulations in *Figure 4—figure supplement 1K–P*, all parameters were identical to the parameter values listed above except for $D$(BMP) = 3 and $D$(Chd) = 6. $D$(Sizzled) was set to 150 in *Figure 4—figure supplement 1K*, and to 10 in *Figure 4—figure supplement 1L–P*. $D$(ADMP) was varied from 0.1 to 150 as indicated in *Figure 4—figure supplement 1K–P*.

## Model 5: Shuttling

For Model 5, a minimal transport model that excludes the effects of downstream patterning circuits was used to illustrate the biophysical aspects of shuttling (*Ben-Zvi et al., 2008*):

$$\frac{\partial[\text{BMP}]}{\partial t} = D_{\text{BMP}}\nabla^2[\text{BMP}] - \kappa[\text{Chd}][\text{BMP}] + \lambda[\text{Xlr}][\text{ChdBMP}] - \lambda_{\text{BMP}}[\text{BMP}] + \rho_{\text{BMP}}(x)$$

$$\frac{\partial[\text{Chd}]}{\partial t} = D_{\text{Chd}}\nabla^2[\text{Chd}] - \kappa[\text{Chd}][\text{BMP}] - \lambda_{\text{Chd}}[\text{Chd}]$$

$$\frac{\partial[\text{ChdBMP}]}{\partial t} = D_{\text{ChdBMP}}\nabla^2[\text{ChdBMP}] + \kappa[\text{Chd}][\text{BMP}] - \lambda[\text{Xlr}][\text{ChdBMP}] - \lambda_{\text{Chd}}[\text{ChdBMP}]$$

## Embryo geometry and boundary conditions

Embryo length: $300 \times 10^{-6}$ m (300 µm)
Constant Chordin flux from the dorsal boundary: $3 \times 10^{-14}$ mol/(m²•s)
No-flux boundary condition for all other species on both ventral and dorsal boundaries

## Parameter values

$D_{BMP}$ = 0.1 µm²/s
$D_{Chd}$ = 10 µm²/s
$D_{ChdBMP}$ = 10 µm²/s
$\lambda_{\text{BMP}}$ = $8.9 \times 10^{-5}$/s (measured in the present study)
$\lambda_{\text{Chd}}$ = $9.6 \times 10^{-5}$/s (measured in the present study)
$\kappa$ = $100 \times 10^3$ m³/(mol•s)
$\lambda = \kappa$
[Xlr] = $2 \times 10^{-8}$ mol/m³
$\rho_{BMP}(x)$ = $0.57 \times 10^{-10} \times e^{-5000x}$ mol/m³ (accounting for the inhomogeneous ventrally peaking distribution of *bmp2b* mRNA in zebrafish embryos)

## Initial conditions

BMP initial concentration: $0.57 \times 10^{-7} \times e^{-5000x}$ mol/m³ throughout the embryo
Chordin initial concentration: 0 mol/m³ everywhere
Chordin-BMP complex initial concentration: 0 mol/m³ everywhere

## Shuttling simulations of adjacent BMP and Chordin clones shown in *Figure 5*

The one-dimensional simulations in *Figure 5* were executed similarly to the ones described above and solved at 15 and 75 min for comparison to the zebrafish embryo double transplantation experiments. The solutions in *Figure 5A* and *Figure 5E* were normalized to the highest free BMP concentration in the simulation without the Chordin source, and the solutions in *Figure 5B* and *Figure 5F* were normalized to the free BMP concentration at the BMP source boundary (at 100 µm) for each condition to facilitate comparison between the gradient ranges.

The double transplantation experiments were modeled using the following equations:

$$\frac{\partial[\text{BMP}]}{\partial t} = D_{\text{BMP}}\nabla^2[\text{BMP}] - \lambda_{\text{BMP}}[\text{BMP}] - \kappa[\text{Chd}][\text{BMP}] + \lambda[\text{Xlr}][\text{ChdBMP}] + \delta_{\text{BMP}}\eta_{\text{BMP}}$$

$$\frac{\partial[\text{Chd}]}{\partial t} = D_{\text{Chd}}\nabla^2[\text{Chd}] - \kappa[\text{Chd}][\text{BMP}] + \delta_{\text{Chd}}\eta_{\text{Chd}}$$

$$\frac{\partial[\text{ChdBMP}]}{\partial t} = D_{\text{ChdBMP}}\nabla^2[\text{ChdBMP}] + \kappa[\text{Chd}][\text{BMP}] - \lambda[\text{Xlr}][\text{ChdBMP}]$$

with

$$\delta_{\text{BMP}} = \begin{cases} 1 \text{ in the BMP source} \\ 0 \text{ otherwise} \end{cases}$$

and

$$\delta_{\text{Chd}} = \begin{cases} 1 \text{ in the Chordin source} \\ 0 \text{ otherwise} \end{cases}$$

## Embryo geometry and boundary conditions

Embryo length: $300 \times 10^{-6}$ m (300 μm)
BMP source: between 50 and 100 μm from the left boundary
Chordin source: between 200 and 250 μm from the left boundary
No-flux boundary conditions on the left and right boundaries

## Parameter values for simulations of shuttling predictions (*Figure 5A+B*)

$D_{BMP}$ = 2 μm$^2$/s (measured in the present study)
$\lambda_{BMP}$ = 0.0001/s (similar to measurements in the present study)
$\eta_{BMP}$ = $5 \times 10^{-5}$ mol/(m$^3$•s)
$\eta_{Chd}$ = $5 \times 10^{-5}$ mol/(m$^3$•s)
$D_{Chd}$ = 100 μm$^2$/s
$D_{ChdBMP}$ = $D_{Chd}$
$\kappa$ = $10 \times 10^3$ m$^3$/(mol•s)
$\lambda$ = $\kappa$
[Xlr] = $2 \times 10^{-7}$ mol/m$^3$

## Parameter values for simulations with experimentally measured diffusivities (*Figure 5E+F*)

$D_{BMP}$ = 2 μm$^2$/s (measured in the present study)
$\lambda_{BMP}$ = 0.0001/s (similar to measurements in the present study)
$\eta_{BMP}$ = $5 \times 10^{-5}$ mol/(m$^3$•s)
$\eta_{Chd}$ = $5 \times 10^{-5}$ mol/(m$^3$•s)
$D_{Chd}$ = 6 μm$^2$/s (measured in the present study)
$D_{ChdBMP}$ = 2.2 μm$^2$/s (measured in the present study)
$\kappa$ = $10 \times 10^3$ m$^3$/(mol s)
$\lambda$ = $\kappa$
[Xlr] = $2 \times 10^{-7}$ mol/m$^3$

## Initial conditions

BMP initial concentration: 0 mol/m$^3$ everywhere
Chordin initial concentration: 0 mol/m$^3$ everywhere
Chordin-BMP complex initial concentration: 0 mol/m$^3$ everywhere

## Acknowledgements

We are grateful to Hans Meinhardt for valuable discussions of BMP/Chordin-mediated dorsal-ventral patterning mechanisms. We thank Edgar Herrera and Luciano Marcon for support with immunostainings, light sheet microscopy, data reconstruction, and helpful discussions. We acknowledge Matteo Pilz and Sarah Keim for technical assistance. This work was supported by the Max Planck Society and a Human Frontier Science Program (HFSP) Career Development Award to PM.

## Additional information

### Funding

| Funder | Grant reference number | Author |
| --- | --- | --- |
| Max-Planck-Gesellschaft | | Patrick Müller |
| Human Frontier Science Program | Career Development Award (CDA00031/2013-C) | Patrick Müller |

The funders had no role in study design, data collection and interpretation, or the decision to submit the work for publication.

## Author contributions
Autumn P Pomreinke, Gary H Soh, Data curation, Formal analysis, Investigation, Visualization, Methodology, Writing—review and editing; Katherine W Rogers, Conceptualization, Resources, Data curation, Formal analysis, Investigation, Visualization, Methodology, Writing—review and editing; Jennifer K Bergmann, Resources, Data curation, Formal analysis, Investigation, Visualization; Alexander J Bläßle, Data curation, Software, Formal analysis, Investigation, Visualization, Methodology, Writing—review and editing; Patrick Müller, Conceptualization, Resources, Data curation, Software, Formal analysis, Supervision, Funding acquisition, Validation, Investigation, Visualization, Methodology, Writing—original draft, Project administration, Writing—review and editing

## Author ORCIDs
Katherine W Rogers, http://orcid.org/0000-0001-5700-2662
Patrick Müller, http://orcid.org/0000-0002-0702-6209

## Decision letter and Author response
Decision letter https://doi.org/10.7554/eLife.25861.024
Author response https://doi.org/10.7554/eLife.25861.025

## Additional files

### Supplementary files
• Transparent reporting form
DOI: https://doi.org/10.7554/eLife.25861.023

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
