## [Decision Letter]

Thank you for submitting your article "Dynamics of BMP signaling and distribution during zebrafish dorsal-ventral patterning" for consideration by *eLife*. Your article has been reviewed by three peer reviewers, and the evaluation has been overseen by Deborah Yelon as the Reviewing Editor and by Didier Stainier as the Senior Editor. The reviewers have opted to remain anonymous.

The reviewers have discussed the reviews with one another and the Reviewing Editor has drafted this decision to help you prepare a revised submission.

Summary:

In this manuscript, Pomreinke et al. examine an intensely debated topic: the mechanism by which BMP and Chordin pattern the dorsal-ventral axis in the early zebrafish embryo. Recent years have seen a proliferation of papers using elaborate models that assume very specific differences between the stabilities and mobilities of ligands and inhibitors, but these parameters are rarely measured directly. Here, the authors perform careful biophysical measurements in order to provide new quantitative information regarding the relevant rates of diffusion and degradation. Their data show that BMP2b and Chordin have similar stabilities and that Chordin is slightly more diffusive. They also show that, in the absence of Chordin, BMP signaling activity is lower on the ventral side but higher dorsally. These are useful observations that represent an advance in determining how the BMP gradient forms. Moreover, this work provides insight into some of the previously proposed models to explain how dorsoventral patterning takes place. Specifically, the authors' data undermine model #4 (shuttling) and provide partial evidence to support model #2 (self-organizing reaction-diffusion system). However, the experimental and theoretical analysis performed here do not yet completely support the authors' claims. Further evidence is needed to strengthen their conclusions, as detailed in the requested Essential Revisions below.

Essential revisions:

1) The authors present four models for the mechanism by which BMP and Chordin pattern the dorsal-ventral axis, but it is not convincing that these fully represent the current understanding in the field. For example, there are also source-sink models that need to be considered.

2) Of the three major conclusions (Discussion, third paragraph), conclusion 1 (Chordin generates signaling peak; reaction-diffusion explains DV patterning) is not convincing, conclusion 2 (BMP does diffuse) is unsurprising, but conclusion 3 (Chordin is more diffusive than BMP) is well supported. In addition, an important fourth conclusion (Chordin does not affect BMP diffusion) should be added.

3) Absolute signaling levels are difficult to measure and compare. How sure are the authors that in the absence of chordin BMP signaling activity is lower on the ventral side?

4) Previous work has shown that chordin mRNA injections at the 1-cell stage rescue zebrafish chordino mutants (Schulte-Merker et al., 1997). Moreover, this brief exposure to uniform Chordin in early embryos is sufficient to generate a normal axial skeleton and fins in rescued mutants, and to grow to adulthood (Fisher and Halpern, 1999). These in vivo observations (of rescue with uniform chordin at early stages) contrast with three theoretical models (Pure inhibition, Long-range accumulation and feedback, Shuttling), which presume a localized Chordin source and point towards the self-organizing system in model #2. It would therefore be valuable for the authors to computationally model how the BMP gradient forms in rescued chordino mutants using parameters that generate uniform Chordin in early embryos. e.g., instead of the assumed/theoretical values in subsections “Simulations of previous models” and “Shuttling simulations of adjacent BMP and Chordin clones shown in Figure 5”, they could test constant and uniform Chordin from dorsal>ventral. Varying the exposure time to uniform Chordin in the simulations would also be informative, i.e., uniform Chordin at early stages vs. uniform early and then localised at the onset of gastrulation, etc.

5) To complement their analysis of bmp/swirl mutants (rescued with bmp2b RNA injections), the authors could experimentally determine by live imaging how the BMP2b-sfGFP gradient forms in chordino mutant embryos upon rescue with chordin RNA injections at the 1-cell stage. Alternatively, measurement of P-Smad levels (as in Figure 1B and Figure 2F) could be sufficient to demonstrate how BMP activity is affected by uniform Chordin.

6) Figure 3: Is BMP2b-sfGFP / Chordin-sfGFP protein expression initially uniform in the injected embryos at late blastula/early gastrula stages (before transplantations to host embryos)? The authors should show this.

7) Results, sixth paragraph: In addition to differences in stability or diffusivity, the production rates of the two proteins could also be different. In Figure 4A+B, it seems that Chordin is detected slightly later than BMP2b. The authors should discuss this.

8) Figure 3—figure supplement Figure 1C: There is a substantial low molecular weight band in the cellular fraction corresponding to cleaved BMP2 (green asterisk in the extracellular enriched fraction) – why is this detected in the cellular fraction? Is there any evidence for intracellular processing of BMP2b?

9) In their test of model 4, "Shuttling": Although the data in Figure 5D does not fit the predicted deflection for shuttling, overall BMP diffusion in Video 10 (with the Chordin clone) seems more pronounced than in Video 9 (control clone). Instead of comparing diffusion of BMP to the right of the clone (towards the Chordin clone) to diffusion coefficients obtained from other embryos, have the authors compared diffusion on the right to BMP diffusion left of the clone (i.e. away from the Chordin clone in the same embryo)?

10) The paper contains important data that are of wide interest, but it is somewhat incomplete because it does not experimentally address Sizzled and ADMP. For example the self-organizing model (subsection “Simulations of previous models”) implies that these proteins have a 25x higher diffusion coefficient than BMP and Chordin. This seems unlikely, because secreted GFP only seems to have a diffusion coefficient that is ~4-6x higher than Chordin.

11) The comparison to Nodal-Lefty is misleading. Lefty has a more than 10x higher diffusion coefficient than Nodal, is much more diffusive than Chordin and is induced by Nodal.

[Editors' note: further revisions were requested prior to acceptance, as described below.]

Thank you for resubmitting your work entitled "Dynamics of BMP signaling and distribution during zebrafish dorsal-ventral patterning" for further consideration at *eLife*. Your revised article has been favorably evaluated by Didier Stainier (Senior Editor), a Reviewing Editor, and two reviewers.

The reviewers agree that the manuscript has been improved, but there remain some issues that need to be addressed before acceptance. The reviewers' concerns are outlined below:

1) It is striking that the authors' conclusions have changed dramatically in the revised manuscript. Notably, the main conclusion in the initial submission now turns out to be wrong ("The inhibitor Chordin is required to establish peak levels of BMP signaling without modulating the diffusivity of BMP2b, challenging shuttling models and providing experimental support for classical reaction-diffusion models of patterning."). Instead, the authors now prefer a graded source-sink model, rather than the self-organising reaction-diffusion model that they preferred in the previous version. Although the current version of the manuscript is improved, the change in conclusions makes one wonder how firm the current model is. It might therefore be wise to revise the paper to make it more data-driven (since the data are convincing and useful) rather than model-driven, and thus leave open the possibility for refined models in the future. Could the authors please revise the presentation of their manuscript accordingly? The emphasis throughout should be on the value of the experimental measurements, rather than on the models, although it is of course appropriate to provide discussion on how well their data fit with particular models.

2) The authors' new results show uniform Chordin-sfGFP distribution through the embryo (Figure 3—figure supplement 1). These results, together with mutant rescue described in the previous reports, suggest that Chordin does not have to be a localised source or sink, since a third of mutant embryos can develop (some even to adulthood) with uniform Chordin. The authors also show that their bmp2b mRNA injections can rescue swirl/bmp2b-/- mutants and express BMP2b-sfGFP protein through the embryo. So, BMP protein does not have to arise from a ventral source either and the rescued mutants can develop with BMP everywhere. The source-sink models are therefore not consistent with these in vivo observations. Altogether, considering the absence of gradient simulation with uniform Chordin, the disconnect between their current model to protein distribution and mutant rescue, and that the models change as additional parameters become available, the graded source-sink model that the authors now prefer cannot be considered conclusive. The authors should therefore discuss the caveats of their current model in light of these observations.

3) Since the authors could not measure ADMP diffusion, did the authors test a range of values in their simulations (instead of making the assumption that ADMP has similar diffusivity to Sizzled)?

---

## [Author Response]

Essential revisions:1) The authors present four models for the mechanism by which BMP and Chordin pattern the dorsal-ventral axis, but it is not convincing that these fully represent the current understanding in the field. For example, there are also source-sink models that need to be considered.

This is an excellent point. We have now included the source-sink model as Model 1 and updated the models with more realistic initial conditions to reflect the non-uniform distribution of *bmp2b* mRNA during early zebrafish development. After a careful assessment of the new data, Model 1 is most consistent with all of our measurements.

2) Of the three major conclusions (Discussion, third paragraph), conclusion 1 (Chordin generates signaling peak; reaction-diffusion explains DV patterning) is not convincing, conclusion 2 (BMP does diffuse) is unsurprising, but conclusion 3 (Chordin is more diffusive than BMP) is well supported. In addition, an important fourth conclusion (Chordin does not affect BMP diffusion) should be added.

We have added this important fourth conclusion to the manuscript as suggested. Regarding conclusion 1, please see point 3 below.

3) Absolute signaling levels are difficult to measure and compare. How sure are the authors that in the absence of *chordin* BMP signaling activity is lower on the ventral side?

The reviewers correctly point out challenges associated with the comparison of absolute fluorescence levels. To minimize artifacts, we took the following precautions: 1) mutant and wild type embryos (Figure 2) were siblings that were fertilized and fixed on the same day, 2) all embryos were processed for immunofluorescence simultaneously using the same reagents, 3) images were acquired in a single imaging session, and 4) gradient quantification was performed before genotypes were established.

To determine whether in the absence of *chordin* BMP signaling activity is lower on the ventral side, we performed an unpaired two-tailed t-test and found that the data sets were not statistically significantly different (p > 0.05 throughout the ventral side, see new Figure 2). We therefore reassessed pSmad1/5/9 immunostainings in a larger number of wild type (n=7) and *chordin* mutants (n=10) and found no consistent statistically significant difference in BMP signaling activity on the ventral side between wild type and *chordin* mutants in any experiment (Figure 2FH). We have adjusted the text and figures accordingly.

4) Previous work has shown that *chordin* mRNA injections at the 1-cell stage rescue zebrafish chordino mutants (Schulte-Merker et al., 1997). Moreover, this brief exposure to uniform Chordin in early embryos is sufficient to generate a normal axial skeleton and fins in rescued mutants, and to grow to adulthood (Fisher and Halpern, 1999). These in vivo observations (of rescue with uniform *chordin* at early stages) contrast with three theoretical models (Pure inhibition, Long-range accumulation and feedback, Shuttling), which presume a localized Chordin source and point towards the self-organizing system in model #2. It would therefore be valuable for the authors to computationally model how the BMP gradient forms in rescued chordino mutants using parameters that generate uniform Chordin in early embryos. e.g., instead of the assumed/theoretical values in subsections “Simulations of previous models” and “Shuttling simulations of adjacent BMP and Chordin clones shown in Figure 5”, they could test constant and uniform Chordin from dorsal>ventral. Varying the exposure time to uniform Chordin in the simulations would also be informative, i.e., uniform Chordin at early stages vs. uniform early and then localised at the onset of gastrulation, etc.

Both points 4 and 5 are extremely interesting ideas that could be studied in the future, but we feel that this analysis is beyond the scope of the paper. We agree that it would be highly valuable to reveal the mechanism by which uniformly provided Chordin can rescue development in *chordin* mutants. Gradient simulations with varying levels of ectopically provided Chordin are feasible, but in the absence of quantitative pSmad1/5/9 data in rescued embryos, such simulations would be at best speculative if not misleading for the field given the almost infinite number of possible Chordin concentrations that could be tested (see point 5 for details). We have therefore decided not to include these simulations in the manuscript.

5) To complement their analysis of bmp/swirl mutants (rescued with bmp2b RNA injections), the authors could experimentally determine by live imaging how the BMP2b-sfGFP gradient forms in chordino mutant embryos upon rescue with *chordin* RNA injections at the 1-cell stage. Alternatively, measurement of P-Smad levels (as in Figure 1B and Figure 2F) could be sufficient to demonstrate how BMP activity is affected by uniform Chordin.

Although it is an excellent suggestion, the first experiment – imaging BMP2b-sfGFP gradient formation in rescued *chordin* mutants – requires BMP2b-sfGFP transgenic lines. However, homologous recombination-mediated large genome modifications are notoriously inefficient in zebrafish, and we have not yet obtained these lines.

To address the second experiment suggestion – measuring pSmad in rescued *chordin* mutants – we first injected *chordin* morphant embryos with Chordin-sfGFP. The *chordin* morpholino is well characterized, highly efficient with minimal off-target effects and toxicity, and targets the 5’UTR of *chordin,* leaving the synthetic Chordin-sfGFP unaffected.However, we observed highly variable rescue of morphants, similar to *chordin* mutant rescue experiments in Fisher and Halpern 1999 (~1/3 show some rescue, ~1/3 are ventralized, and ~1/3 are dorsalized at 24 hpf). It would be useful to measure the pSmad1/5/9 distributions in fully rescued embryos at shield stage, but since it is not clear what phenotype the fixed embryos would have displayed later at around 24 hpf given the rescue variability, it is challenging to establish a clear link between Chordin distributions, pSmad1/5/9 gradients, and phenotypic rescue. It might be possible to establish a correlation by analyzing large numbers of embryos (i.e., hundreds), but this large-scale approach is incompatible with our low-throughput quantitative immunostaining measurements of pSmad1/5/9 distributions. We have generated light sheet images of pSmad1/5/9 and ChordinsfGFP in *chordin* morphants, but we chose not to include this inconclusive data in our study given the conceptual problems with data interpretation outlined above.

6) Figure 3: Is BMP2b-sfGFP / Chordin-sfGFP protein expression initially uniform in the injected embryos at late blastula/early gastrula stages (before transplantations to host embryos)? The authors should show this.

We now show in Figure 3—figure supplement 1 that Chordin-sfGFP is distributed nearly uniformly, whereas BMP2b-sfGFP distribution tends to be regionally biased in donor embryos, likely due to injection inhomogeneities combined with the lower diffusivity of BMP2b-sfGFP. But even with non-uniform expression across the donor embryo, our conclusions would remain unaffected since small clones of cells transplanted from donors to hosts likely provide a constant relative flux of protein irrespective of the absolute protein concentration or distribution in the donor.

7) Results, sixth paragraph: In addition to differences in stability or diffusivity, the production rates of the two proteins could also be different. In Figure 4A+B, it seems that Chordin is detected slightly later than BMP2b. The authors should discuss this.

The production rates could indeed be different, and this would affect the amplitude of the gradients but not their shape or range (Wartlick et al. 2009). In Figure 3 we therefore normalized the gradients to the concentration at the source boundary, i.e. gradient amplitude, to highlight the kinetics of gradient shape evolution over time. We now include an additional clarification in the Materials and methods (light sheet microscopy section): “After subtracting this value from the experimental intensity profiles, the data was normalized to the value closest to the clonal source boundary. This approach allows for the comparison of the relative gradient range, which is independent of constant production rates. We assume constant production rates over the relatively short time scales of observation (≈ 80 min)”.

The experiment in Figure 4 does not contain information about production rates. For technical reasons, the experiments for Chordin-Dendra2 were on average started slightly later after photoconversion than the experiments for BMP2b-Dendra2, resulting in the slight time shift between the curves.

8) Figure 3—figure supplement 1C: There is a substantial low molecular weight band in the cellular fraction corresponding to cleaved BMP2 (green asterisk in the extracellular enriched fraction) – why is this detected in the cellular fraction? Is there any evidence for intracellular processing of BMP2b?

Our centrifugation method does not fully separate extracellular and cellular fractions, and some of the supernatant (extracellular enriched fraction) remains with the pellet (cellular fraction), which could explain residual mature BMP2b in the cellular fraction. Moreover, it is likely that some BMP2b molecules remain bound to extracellular matrix components associated with the cellular fraction. Binding interactions with immobile membrane-associated molecules could be relevant and might explain the differences between the hindered/global and the free/local diffusion coefficient of BMP2b (Figure 4, Müller et al. 2013).

Regarding the intracellular processing of BMP2b, there is a debate in the field about whether TGFβ superfamily ligands are intracellularly or extracellularly cleaved (reviewed in Constam 2014). There is evidence that some BMPs are cleaved intracellularly (Degnin et al., 2004), but processing may vary between different BMP ligands (Fritsch et al. 2012).

9) In their test of model 4, "Shuttling": Although the data in Figure 5D does not fit the predicted deflection for shuttling, overall BMP diffusion in Video 10 (with the Chordin clone) seems more pronounced than in Video 9 (control clone). Instead of comparing diffusion of BMP to the right of the clone (towards the Chordin clone) to diffusion coefficients obtained from other embryos, have the authors compared diffusion on the right to BMP diffusion left of the clone (i.e. away from the Chordin clone in the same embryo)?

We did not measure diffusivity in embryos that received BMP2b-sfGFP or Chordin-sfGFP-expressing clones. Instead, to specifically address whether Chordin affects BMP2b diffusivity, we performed FRAP in embryos uniformly overexpressing BMP2b-sfGFP only, and in embryos uniformly overexpressing both BMP2b-sfGFP and Chordin (Figure 4). These relatively straightforward experiments demonstrated a non-significant effect of Chordin on BMP diffusivity. Measuring BMP diffusivity in embryos receiving transplanted clones is more complicated and would require more inferences than our FRAP experiments. For example, we would have to infer Chordin concentration as a function of distance from the Chordin clone. We therefore argue that our FRAP experiments are a cleaner and more robust methodology to address this question.

We failed to observe consistent differences in BMP2b-sfGFP distribution in clonal transplantation experiments in the presence or absence of Chordin-secreting clones. We now provide additional videos of BMP2b gradient formation to clarify this point (Video 9 and Video 10).

10) The paper contains important data that are of wide interest, but it is somewhat incomplete because it does not experimentally address Sizzled and ADMP. For example the self-organizing model (subsection “Simulations of previous models”) implies that these proteins have a 25x higher diffusion coefficient than BMP and Chordin. This seems unlikely, because secreted GFP only seems to have a diffusion coefficient that is ~4-6x higher than Chordin.

We thank the reviewers for raising this important point. We have extended our biophysical analysis to include Sizzled and now provide a new figure (Figure 4—figure supplement 1) dedicated to the characterization of Sizzled diffusion and its role in BMP signal gradient formation. We also attempted to do the same for ADMP; unfortunately, although we generated three different ADMP fusion proteins (sfGFP inserted 2, 5, and 11 amino acids downstream of the Furin cleavage site with different linkers), we did not succeed in generating a construct that had similar activity as untagged ADMP. We therefore do not include a biophysical characterization of ADMP in this paper.

We found that Sizzled is slightly more diffusive than BMP, but not by a factor of 25 as correctly speculated by the reviewers. We additionally simulated the models (Models 3 and 4) that depend on ADMP and Sizzled for BMP signal gradient formation using our Sizzled FRAP measurements. While Model 3 can form dorsal-ventral gradients with these realistic values, the gradient formation kinetics are not consistent with our measurements (Figure 4—figure supplement 1). Strikingly, Model 4 fails to form any gradient with the diffusion coefficient ratios that we measured (and assuming an ADMP diffusivity similar to that of Sizzled, Figure 4—figure supplement 1), showing that the model in its current form cannot accurately account for the kinetics of dorsal-ventral patterning.

11) The comparison to Nodal-Lefty is misleading. Lefty has a more than 10x higher diffusion coefficient than Nodal, is much more diffusive than Chordin and is induced by Nodal.

We removed this section from the manuscript.

[Editors' note: further revisions were requested prior to acceptance, as described below.]

The reviewers agree that the manuscript has been improved, but there remain some issues that need to be addressed before acceptance. The reviewers' concerns are outlined below:1) It is striking that the authors' conclusions have changed dramatically in the revised manuscript. Notably, the main conclusion in the initial submission now turns out to be wrong ("The inhibitor Chordin is required to establish peak levels of BMP signaling without modulating the diffusivity of BMP2b, challenging shuttling models and providing experimental support for classical reaction-diffusion models of patterning."). Instead, the authors now prefer a graded source-sink model, rather than the self-organising reaction-diffusion model that they preferred in the previous version. Although the current version of the manuscript is improved, the change in conclusions makes one wonder how firm the current model is. It might therefore be wise to revise the paper to make it more data-driven (since the data are convincing and useful) rather than model-driven, and thus leave open the possibility for refined models in the future. Could the authors please revise the presentation of their manuscript accordingly? The emphasis throughout should be on the value of the experimental measurements, rather than on the models, although it is of course appropriate to provide discussion on how well their data fit with particular models.

The goal of our study was to test current BMP/Chordin-mediated dorsal-ventral patterning models by experimentally assessing as many of their predictions as possible. This unbiased approach allowed us to rule out some models, while providing support for others. Based on the reviewers’ input, we tested an additional model, performed new experiments and simulations, acquired further information about the key players in this system, and performed rigorous statistical tests on all of our data to strengthen the conclusions of the paper. This resulted in new experimental findings that were inconsistent with the models that we previously tested (Models 2-5) but that were consistent with the additional model suggested by the reviewers (Model 1). We agree that this model will likely require refinement in the future as more data becomes available, and have now added a more extensive discussion of caveats and unanswered questions regarding the mechanisms of BMP gradient formation as suggested.

2) The authors' new results show uniform Chordin-sfGFP distribution through the embryo (Figure 3—figure supplement 1). These results, together with mutant rescue described in the previous reports, suggest that Chordin does not have to be a localised source or sink, since a third of mutant embryos can develop (some even to adulthood) with uniform Chordin. The authors also show that their bmp2b mRNA injections can rescue swirl/bmp2b-/- mutants and express BMP2b-sfGFP protein through the embryo. So, BMP protein does not have to arise from a ventral source either and the rescued mutants can develop with BMP everywhere. The source-sink models are therefore not consistent with these in vivo observations. Altogether, considering the absence of gradient simulation with uniform Chordin, the disconnect between their current model to protein distribution and mutant rescue, and that the models change as additional parameters become available, the graded source-sink model that the authors now prefer cannot be considered conclusive. The authors should therefore discuss the caveats of their current model in light of these observations.

Based on the rescue of mutants with ectopic BMP or Chordin, it is possible that normal patterning could occur as long as one component of the system is localized (i.e. ventrally biased *bmp* expression with uniform Chordin, or dorsally biased *chordin* expression with uniform BMP) leading to a signaling bias. To determine whether the models under investigation are consistent with mutant rescue, it would be highly relevant to assess the pSmad1/5/9 distribution in rescued mutants, and ideally assess the distribution of the ectopic protein as well. This would address some assumptions about rescue: namely, that BMP signaling is correctly recapitulated during early development, and that ectopic protein is actually uniformly distributed. For example, expression of injected mRNA is not always uniform, and the 30% of embryos that are rescued may have an expression bias. We attempted to answer these questions experimentally, but were unable to obtain interpretable results due to technical limitations (see our previous response to the reviewers’ comments). Unfortunately, reasonable models cannot currently be developed without measurements of BMP signaling and protein distribution under rescuing conditions. To address these limitations as suggested by the reviewers, our manuscript now includes a discussion about the caveats of the currently best-supported model, including the fact that apparently uniform Chordin/BMP expression can rescue mutants.

3) Since the authors could not measure ADMP diffusion, did the authors test a range of values in their simulations (instead of making the assumption that ADMP has similar diffusivity to Sizzled)?

In the revised manuscript we now provide additional simulations of Models 3 and 4 using six biologically plausible ADMP diffusion coefficients spanning three orders of magnitude. None of the tested conditions produce reasonable dorsal-ventral BMP signaling gradients when the diffusivities of BMP, Chordin, and Sizzled are constrained with the experimentally measured values.